# Learning with Fitzpatrick Losses

**Seta Rakotomandimby**
Ecole des Ponts
seta.rakotomandimby@enpc.fr

**Jean-Philippe Chancelier**
Ecole des Ponts
jean-philippe.chancelier@enpc.fr

**Michel De Lara**
Ecole des Ponts
michel.delara@enpc.fr

**Mathieu Blondel**
Google DeepMind
mblondel@google.com

## Abstract

Fenchel-Young losses are a family of loss functions, encompassing the squared, logistic and sparsemax losses, among others. They are convex w.r.t. the model output and the target, separately. Each Fenchel-Young loss is implicitly associated with a link function, that maps model outputs to predictions. For instance, the logistic loss is associated with the soft argmax link function. Can we build new loss functions associated with the same link function as Fenchel-Young losses? In this paper, we introduce Fitzpatrick losses, a new family of separately convex loss functions based on the Fitzpatrick function. A well-known theoretical tool in maximal monotone operator theory, the Fitzpatrick function naturally leads to a refined Fenchel-Young inequality, making Fitzpatrick losses tighter than Fenchel-Young losses, while maintaining the same link function for prediction. As an example, we introduce the Fitzpatrick logistic loss and the Fitzpatrick sparsemax loss, counterparts of the logistic and the sparsemax losses. This yields two new tighter losses associated with the soft argmax and the sparse argmax, two of the most ubiquitous output layers used in machine learning. We study in details the properties of Fitzpatrick losses and, in particular, we show that they can be seen as Fenchel-Young losses using a modified, target-dependent generating function. We demonstrate the effectiveness of Fitzpatrick losses for label proportion estimation.

## 1 Introduction

Loss functions are a cornerstone of statistics and machine learning: they measure the difference, or "loss," between a ground-truth target and a model prediction. As such, they have attracted a wealth of research. Proper losses (a.k.a. proper scoring rules) [17, 16] measure the discrepancy between a target distribution and a probability forecast. They are essentially primal-primal Bregman divergences, with both the target and the prediction belonging to the same primal space. They are typically explicitly composed with a link function [26, 30], in order to map the model output to a prediction. A disadvantage of this explicit composition is that it often makes the resulting composite loss function non-convex. A related family of loss functions are Fenchel-Young losses [7, 8], which encompass many commonly-used loss functions in machine learning including the squared, logistic, sparsemax and perceptron losses. Fenchel-Young losses can be seen as primal-dual Bregman divergences [1], with the target belonging to the primal space and the model output belonging to the dual space. In contrast to proper losses, each Fenchel-Young loss is implicitly associated with a given link function, mapping the dual-space model output to a primal-space prediction (for instance, the soft argmax is the link function associated with the logistic loss). This crucial difference makes Fenchel-Young losses always convex w.r.t. the model output and w.r.t. the target, separately. Can we build new (separately) convex losses associated with the same link function as Fenchel-Young losses?

38th Conference on Neural Information Processing Systems (NeurIPS 2024).

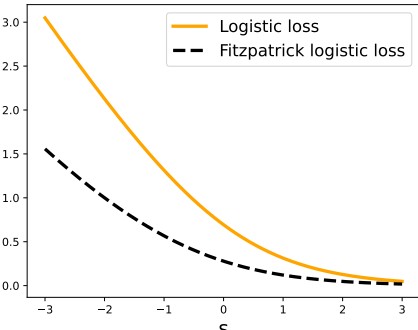 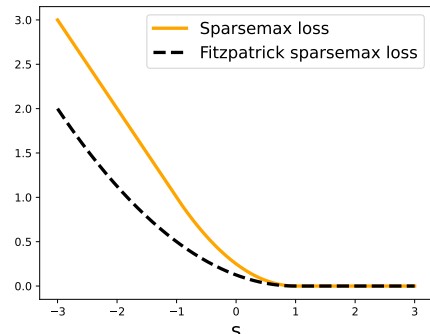

Figure 1: We introduce **Fitzpatrick losses**, a new family of loss functions $L(y, \theta)$ generated by a convex regularization function $\Omega$, that are **lower-bounds** of the Fenchel-Young losses generated by the same function $\Omega$, while maintaining the **same** link function $\widehat{y}_\Omega = \nabla\Omega^*$. In particular, we use our framework to instantiate the counterparts of the **logistic** and **sparsemax** losses, two instances of Fenchel-Young losses, associated with the **soft argmax** and the **sparse argmax**. In the figures above, we plot $L(y, \theta)$, where $y = e_1$, $\theta = (s, 0)$ and $L \in \{L_{F[\partial\Omega]}, L_{\Omega \oplus \Omega^*}\}$, confirming the lower-bound property.

In this paper, we introduce Fitzpatrick losses, a new family of primal-dual (separately) convex loss functions. Our proposal builds upon the Fitzpatrick function, a well-known theoretical object in maximal monotone operator theory [15, 11, 2]. So far, the Fitzpatrick function had been used as a theoretical tool to represent maximal monotone operators [28] and to construct Bregman-like primal-primal divergences [10], but it had not been used to construct primal-dual loss functions for machine learning, as we do. Crucially, the Fitzpatrick function naturally leads to a refined Fenchel-Young inequality, making Fitzpatrick losses tighter than Fenchel-Young losses. Yet, their predictions are produced using the same link function, suggesting that we can use Fitzpatrick losses as a tighter replacement for the corresponding Fenchel-Young losses (Figure 1). We make the following contributions.

- After reviewing some background, we introduce Fitzpatrick losses. They can be thought as a tighter version of Fenchel-Young losses, that use the same link function.

- We instantiate two new loss functions in this family: the Fitzpatrick logistic loss and the Fitzpatrick sparsemax loss. They are the counterparts of the logistic and sparsemax losses, two instances of Fenchel-Young losses. We therefore obtain two new tighter losses for the soft argmax and the sparse argmax, two of the most popular output layers in machine learning.

- We study in detail the properties of Fitzpatrick losses. We show that Fitzpatrick losses are equivalent to Fenchel-Young losses with a modified, target-dependent generating function.

- We demonstrate the effectiveness of Fitzpatrick losses for probabilistic classification on 11 datasets.

## 2 Background

### 2.1 Convex analysis

We denote the non-negative real numbers by $\mathbb{R}_+ := [0, +\infty)$, the positive real numbers by $\mathbb{R}_{++} := (0, +\infty)$, and the extended real numbers by $\overline{\mathbb{R}} := \mathbb{R} \cup \{-\infty, +\infty\}$. We suppose given a nonzero natural number $k$. We denote the probability simplex by $\triangle^k := \{p \in \mathbb{R}_+^k \mid \sum_{i=1}^k p_i = 1\}$. We denote the indicator function of a set $\mathcal{C} \subset \mathbb{R}^k$ by $\iota_{\mathcal{C}}(y) = 0$ if $y \in \mathcal{C}$, $+\infty$ otherwise. We denote the effective domain of a function $\Omega : \mathbb{R}^k \to \overline{\mathbb{R}}$ by $\operatorname{dom}\Omega := \{y \in \mathbb{R}^k \mid \Omega(y) < +\infty\}$. A function $\Omega : \mathbb{R}^k \to \overline{\mathbb{R}}$ is said to be **proper** if it never takes the value $-\infty$ and if $\operatorname{dom}\Omega \neq \emptyset$. We denote the Euclidean projection onto a nonempty closed convex set $\mathcal{C} \subset \mathbb{R}^k$ by $P_{\mathcal{C}}(\theta)$, the unique solution $y' \in \mathcal{C}$ of the minimization problem $\min_{y' \in \mathcal{C}} \|y' - \theta\|_2^2$.

For a function $\Omega : \mathbb{R}^k \to \overline{\mathbb{R}}$, its **subdifferential** $\partial\Omega \subset \mathbb{R}^k \times \mathbb{R}^k$ is defined by

$$(y', \theta') \in \partial\Omega \iff \theta' \in \partial\Omega(y') \iff \Omega(y) \geq \Omega(y') + \langle y - y', \theta' \rangle, \; \forall y \in \mathbb{R}^k.$$

When $\Omega$ is convex and differentiable, the subdifferential is a singleton and we have $\partial\Omega(y') = \{\nabla\Omega(y')\}$. The **normal cone** to a nonempty closed convex set $\mathcal{C} \subset \mathbb{R}^k$ at $y'$ is defined by

$$\theta' \in N_{\mathcal{C}}(y') \iff \langle y - y', \theta' \rangle \leq 0, \quad \forall y \in \mathcal{C}$$

if $y' \in \mathcal{C}$ and $N_{\mathcal{C}}(y') = \emptyset$ if $y' \notin \mathcal{C}$. The **Fenchel conjugate** $\Omega^* : \mathbb{R}^k \to \overline{\mathbb{R}}$ of a function $\Omega : \mathbb{R}^k \to \overline{\mathbb{R}}$ is defined by

$$\Omega^*(\theta) := \sup_{y' \in \mathbb{R}^k} \langle y', \theta \rangle - \Omega(y').$$

From standard results in convex analysis [27, Proposition 11.3], when $\Omega : \mathbb{R}^k \to \overline{\mathbb{R}}$ is a proper convex l.s.c. (lower semicontinuous) function,

$$\partial\Omega^*(\theta) = \operatorname*{argmax}_{y' \in \mathbb{R}^k} \langle y', \theta \rangle - \Omega(y').$$

When the argmax is unique, it is equal to $\nabla\Omega^*(\theta)$. We define the **generalized Bregman divergence** [18] $D_\Omega : \mathbb{R}^k \times \mathbb{R}^k \to \overline{\mathbb{R}}_+$ generated by a proper convex l.s.c. function $\Omega : \mathbb{R}^k \to \overline{\mathbb{R}}$, by

$$D_\Omega(y, y') := \Omega(y) - \Omega(y') - \sup_{\theta' \in \partial\Omega(y')} \langle y - y', \theta' \rangle, \tag{1}$$

with the convention $+\infty + (-\infty) = +\infty$ (we comment on this convention in Appendix C). When $\Omega$ is differentiable, Equation (1) gives the classical **Bregman divergence**

$$D_\Omega(y, y') := \Omega(y) - \Omega(y') - \langle y - y', \nabla\Omega(y') \rangle.$$

Both $y$ and $y'$ belong to the **primal space**.

## 2.2  Fenchel-Young losses

### Definition and properties

The **Fenchel-Young loss** [8] $L_{\Omega \oplus \Omega^*} : \mathbb{R}^k \times \mathbb{R}^k \to \overline{\mathbb{R}}$ generated by a proper convex l.s.c. function $\Omega$ is

$$L_{\Omega \oplus \Omega^*}(y, \theta) := \left(\Omega \oplus \Omega^*\right)(y, \theta) - \langle y, \theta \rangle := \Omega(y) + \Omega^*(\theta) - \langle y, \theta \rangle.$$

As its name indicates, it is grounded in the Fenchel-Young inequality

$$\langle y, \theta \rangle \leq \Omega(y) + \Omega^*(\theta) \quad \forall y, \theta \in \mathbb{R}^k.$$

The Fenchel-Young loss enjoys many desirable properties, notably it is **non-negative** and it is **separately convex** in $y$ and $\theta$. The Fenchel-Young loss can be seen as a **primal-dual** Bregman divergence [1, 8], where $y$ belongs to the primal space and $\theta$ belongs to the dual space.

### Link functions

To map an element $\theta$ of a dual space to an element $y$ of a primal space, we define the link function (potentially set-valued) associated with the loss $L$ by

$$\theta \mapsto \{y \in \mathbb{R}^k \mid L(y, \theta) = 0\}.$$

Given a proper convex function $\Omega$, the associated Fenchel-Young loss $L_{\Omega \oplus \Omega^*}$ produces the canonical link function $\partial\Omega^*$, since

$$L_{\Omega \oplus \Omega^*}(y, \theta) = 0 \iff y \in \partial\Omega^*(\theta).$$

In particular when $\Omega$ is strictly convex, and thus $\Omega^*$ is differentiable according to [27, Theorem 11.13], the Fenchel-Young loss satisfies the identity of indiscernibles

$$L_{\Omega \oplus \Omega^*}(y, \theta) = 0 \iff y = \nabla\Omega^*(\theta).$$

In the remainder of this paper, we will use the notation $\widehat{y}_\Omega(\theta)$ for the canonical link function $\partial\Omega^*(\theta)$. When the function $\Omega^*$ is differentiable, $\widehat{y}_\Omega(\theta)$ will denote $\nabla\Omega^*(\theta)$. Since $\Omega^*$ is convex, $\widehat{y}_\Omega$ is monotone (see §2.3). As shown in [8], the monotonicity implies that $\theta$ and $\widehat{y}_\Omega(\theta)$ are sorted the same way, i.e., $\theta_i > \theta_j \implies \widehat{y}_\Omega(\theta)_i \geq \widehat{y}_\Omega(\theta)_j$. Link functions also play an important role in the loss subgradients (and in the loss gradient when it is differentiable), as we have

$$\partial_\theta L_{\Omega \oplus \Omega^*}(y, \theta) = \widehat{y}_\Omega(\theta) - y. \tag{2}$$

**Examples of Fenchel-Young loss instances and their associated link function**

We give a few examples of Fenchel-Young losses. With the squared 2-norm, $\Omega(y') = \frac{1}{2}\|y'\|_2^2$, we obtain the **squared loss**

$$L_{\Omega \oplus \Omega^*}(y, \theta) = L_{\text{squared}}(y, \theta) := \frac{1}{2}\|y - \theta\|_2^2$$

and the **identity link**

$$\widehat{y}_\Omega(\theta) = \theta.$$

With the indicator of a nonempty closed convex set $\mathcal{C}$, $\Omega(y') = \iota_{\mathcal{C}}(y')$, we obtain the **perceptron loss**

$$L_{\Omega \oplus \Omega^*}(y, \theta) = L_{\text{perceptron}}(y, \theta) := \max_{y' \in \mathcal{C}} \ \langle y', \theta \rangle - \langle y, \theta \rangle$$

and the **argmax link**

$$\widehat{y}_\Omega(\theta) = \underset{y \in \mathcal{C}}{\operatorname{argmax}} \ \langle y, \theta \rangle.$$

With the squared 2-norm restricted to some nonempty closed convex set $\mathcal{C}$, $\Omega(y') = \frac{1}{2}\|y'\|_2^2 + \iota_{\mathcal{C}}(y')$, we obtain the **sparseMAP loss** [24]

$$L_{\Omega \oplus \Omega^*}(y, \theta) = L_{\text{sparseMAP}}(y, \theta) := \frac{1}{2}\|y - \theta\|_2^2 - \frac{1}{2}\|P_{\mathcal{C}}(\theta) - \theta\|_2^2,$$

and the link is the **Euclidean projection** onto $\mathcal{C}$,

$$\widehat{y}_\Omega(\theta) = P_{\mathcal{C}}(\theta).$$

When the set is $\mathcal{C} = \triangle^k$, we obtain the **sparsemax loss** [21] and the **sparse argmax link** $\widehat{y}_\Omega(\theta) = P_{\triangle^k}(\theta)$ (also known as sparsemax), which is known to produce sparse probability distributions.

With the Shannon negentropy restricted to the probability simplex, $\Omega(y') := \langle y', \log y' \rangle + \iota_{\triangle^k}(y')$, we obtain the **logistic loss**

$$L_{\Omega \oplus \Omega^*}(y, \theta) = L_{\text{logistic}}(y, \theta) := \log \sum_{i=1}^{k} \exp(\theta_i) + \langle y, \log y \rangle - \langle y, \theta \rangle,$$

and the **soft argmax link** (also known as softmax)

$$\widehat{y}_\Omega(\theta) = \operatorname{softargmax}(\theta) := \exp(\theta) / \sum_{i=1}^{k} \exp(\theta_i).$$

## 2.3 Maximal monotone operators and the Fitzpatrick function

An operator $A$, that is, a subset $A \subset \mathbb{R}^k \times \mathbb{R}^k$, is called **monotone** if for all $(y, \theta) \in A$ and all $(y', \theta') \in A$, we have

$$\langle y' - y, \theta' - \theta \rangle \geq 0.$$

We overload the notation to denote $A(y) := \{\theta \in \mathbb{R}^k \mid (y, \theta) \in A\}$. A monotone operator $A$ is said to be **maximal** if there does not exist $(y, \theta) \notin A$ such that $A \cup \{(y, \theta)\}$ is still monotone. It is well-known that the subdifferential $\partial \Omega$ of a proper convex l.s.c. function $\Omega$ is maximal monotone. For more details on monotone operators, see [3, 28].

A well-known object in monotone operator theory, the **Fitzpatrick function** associated with a maximal monotone operator $A$ [15, 11, 2], denoted $F[A] : \mathbb{R}^k \times \mathbb{R}^k \to \overline{\mathbb{R}}$, is defined by

$$F[A](y, \theta) := \sup_{(y', \theta') \in A} \langle y - y', \theta' \rangle + \langle y', \theta \rangle.$$

In particular, with $A = \partial \Omega$, we have

$$F[\partial \Omega](y, \theta) = \sup_{(y', \theta') \in \partial \Omega} \langle y - y', \theta' \rangle + \langle y', \theta \rangle = \sup_{y' \in \operatorname{dom} \Omega} \left[ \langle y', \theta \rangle + \sup_{\theta' \in \partial \Omega(y')} \langle y - y', \theta' \rangle \right].$$

The Fitzpatrick function was studied in depth in [2]. In particular, it is (jointly) convex and satisfies

$$\langle y, \theta \rangle \leq F[\partial\Omega](y, \theta) \leq \left(\Omega \oplus \Omega^*\right)(y, \theta) = \Omega(y) + \Omega^*(\theta) \quad \forall y, \theta \in \mathbb{R}^k. \tag{3}$$

We introduce the operator $y^{\star}_{F[\partial\Omega]} \subset (\mathbb{R}^k \times \mathbb{R}^k) \times \mathbb{R}^k$, associated to the Fitzpatrick function $F[\partial\Omega]$:

$$y^{\star}_{F[\partial\Omega]}(y, \theta) := \operatorname*{argmax}_{y' \in \mathrm{dom}\,\Omega} \left[ \langle y', \theta \rangle + \sup_{\theta' \in \partial\Omega(y')} \langle y - y', \theta' \rangle \right]. \tag{4}$$

As proven for Item 4 in Proposition 1, we have $y^{\star}_{F[\partial\Omega]}(y, \theta) \subset \partial_\theta F[\partial\Omega](y, \theta)$. For the rest of the paper, in case that the $\operatorname{argmax}$ in (4) is a singleton $\{y^\star\}$, we will write $y^{\star}_{F[\partial\Omega]}(y, \theta) := y^\star$. The Fitzpatrick function $F[\partial\Omega](y, \theta)$ and $\left(\Omega \oplus \Omega^*\right)(y, \theta) = \Omega(y) + \Omega^*(\theta)$ play a similar role but the latter function is **separable** in $y$ and $\theta$, while the former is **not**. In particular this makes the subdifferential $\partial_\theta F[\partial\Omega](y, \theta)$ depend on both $y$ and $\theta$, while $\partial_\theta(\Omega \oplus \Omega^*)(y, \theta) = \partial\Omega^*(\theta)$ depends only on $\theta$.

The Fitzpatrick function was used in [10] to theoretically study primal-primal Bregman-like divergences. As discussed in more detail in Section 3.4, using these divergences for machine learning would require us to compose them with an explicit link function, which would typically break convexity. In the next section, we introduce new primal-dual losses based on the Fitzpatrick function.

## 3 Fitzpatrick losses

### 3.1 Definition and properties

Inspired by the inequality in (3), which we can view as a refined Fenchel-Young inequality, we introduce Fitzpatrick losses, a new family of loss functions generated by a convex l.s.c. function $\Omega$.

---

**Definition 1** *Fitzpatrick loss*

*Let $\Omega : \mathbb{R}^k \to \overline{\mathbb{R}}$ be a proper convex l.s.c. function. When $y \in \mathrm{dom}\,\Omega$ and $\theta \in \mathbb{R}^k$, we define the Fitzpatrick loss $L_{F[\partial\Omega]} : \mathbb{R}^k \times \mathbb{R}^k \to \overline{\mathbb{R}}$ generated by $\Omega$ as*

$$\begin{aligned} L_{F[\partial\Omega]}(y, \theta) &:= F[\partial\Omega](y, \theta) - \langle y, \theta \rangle \\ &= \sup_{(y', \theta') \in \partial\Omega} \langle y - y', \theta' \rangle + \langle y', \theta \rangle - \langle y, \theta \rangle \\ &= \sup_{(y', \theta') \in \partial\Omega} \langle y' - y, \theta - \theta' \rangle. \end{aligned}$$

*When $y \notin \mathrm{dom}\,\Omega$, $L_{F[\partial\Omega]}(y, \theta) := +\infty$.*

---

Fitzpatrick losses enjoy similar properties as Fenchel-Young losses, while being **tighter** than Fenchel-Young losses.

---

**Proposition 1** *Properties of Fitzpatrick losses*

1. **Non-negativity:** *for all $(y, \theta) \in \mathbb{R}^k \times \mathbb{R}^k$, $L_{F[\partial\Omega]}(y, \theta) \geq 0$.*

2. **Same link function:** $L_{\Omega \oplus \Omega^*}(y, \theta) = L_{F[\partial\Omega]}(y, \theta) = 0 \iff y \in \widehat{y}_\Omega(\theta)$.

3. **Separable convexity:** $L_{F[\partial\Omega]}(y, \theta)$ *is separately convex.*

4. **(Sub-)Gradient:** $\partial_\theta L_{F[\partial\Omega]}(y, \theta) \supset y^{\star}_{F[\partial\Omega]}(y, \theta) - y$ *where $y^{\star}_{F[\partial\Omega]}(y, \theta)$ is given by (4).*

5. **Tighter inequality:** *for all $(y, \theta) \in \mathbb{R}^k$, $0 \leq L_{F[\partial\Omega]}(y, \theta) \leq L_{\Omega \oplus \Omega^*}(y, \theta)$.*

---

A proof is given in Appendix B.2. Because the Fitzpatrick loss and the Fenchel-Young loss generated by the same $\Omega$ have the same link function $\widehat{y}_\Omega$, they share the same minimizers w.r.t. $\theta$ for $y$ fixed. However, the Fitzpatrick loss is always a **lower bound** of the corresponding Fenchel-Young loss. Moreover, they have different gradients w.r.t. $\theta$: $\partial_\theta L_{\Omega \oplus \Omega^*}(y, \theta) = \widehat{y}_\Omega(\theta) - y$ vs. $\partial_\theta L_{F[\partial\Omega]}(y, \theta) \supset y^{\star}_{F[\partial\Omega]}(y, \theta) - y$. It is worth noticing that $y^{\star}_{F[\partial\Omega]}(y, \theta)$ depends on both $y$ and $\theta$, contrary to $\widehat{y}_\Omega(\theta)$.

When $\Omega$ is a twice differentiable function on its domain (which is for instance the case of the squared 2-norm or the negentropy), we next show that Fitzpatrick losses enjoy a particularly simple expression and become a squared Mahalanobis-like distance.

---

**Proposition 2** *Expressions of $F[\partial\Omega](y,\theta)$ and $L_{F[\partial\Omega]}(y,\theta)$ when $\Omega$ is twice differentiable*

*Let $\Omega : \mathbb{R}^k \to \overline{\mathbb{R}}$ be a convex function such that* dom $\Omega$ *is an open set. Let us assume that $\Omega$ is twice differentiable. Then, for any $y \in \operatorname{dom}\Omega$ and for any $y^\star \in y^\star_{F[\partial\Omega]}(y,\theta)$, as defined in (4), we have that*

$$F[\partial\Omega](y,\theta) = \langle y, \nabla\Omega(y^\star)\rangle + \langle y^\star, \theta\rangle - \langle y^\star, \nabla\Omega(y^\star)\rangle$$
$$L_{F[\partial\Omega]}(y,\theta) = \langle y^\star - y, \theta - \nabla\Omega(y^\star)\rangle$$
$$= \langle y^\star - y, \nabla^2\Omega(y^\star)(y^\star - y)\rangle$$

*and*

$$\nabla^2\Omega(y^\star)(y^\star - y) = \theta - \nabla\Omega(y^\star).$$

---

A proof is given in B.3. When $\Omega$ is constrained (i.e., when it contains an indicator function), we show in §3.5 that the above expression becomes a lower bound.

## 3.2   Examples

We now present the Fitzpatrick loss counterparts of various Fenchel-Young losses.

**Squared loss.**

---

**Proposition 3** *Squared loss as a Fitzpatrick loss*

*When $\Omega(y') = \frac{1}{2}\|y'\|_2^2$, we have for all $y \in \mathbb{R}^k$ and $\theta \in \mathbb{R}^k$*

$$L_{F[\partial\Omega]}(y,\theta) = \frac{1}{4}\|y - \theta\|_2^2 = \frac{1}{2}L_{\text{squared}}(y,\theta).$$

---

A proof is given in Appendix B.4. Therefore, the Fenchel-Young and Fitzpatrick losses generated by $\Omega$ coincide, but up to a factor $\frac{1}{2}$.

**Perceptron loss.**

---

**Proposition 4** *Perceptron loss as a Fitzpatrick loss*

*When $\Omega(y') = \iota_{\mathcal{C}}(y')$, where $\mathcal{C}$ is a nonempty closed convex set, we have for all $y \in \mathcal{C}$ and $\theta \in \mathbb{R}^k$*

$$L_{F[\partial\Omega]}(y,\theta) = L_{\text{perceptron}}(y,\theta) = \max_{y' \in \mathcal{C}} \langle y', \theta\rangle - \langle y, \theta\rangle.$$

---

A proof is given in Appendix B.5. Therefore, the Fenchel-Young and Fitzpatrick losses generated by $\Omega$ exactly coincide in this case.

**Fitzpatrick sparseMAP and Fitzpatrick sparsemax losses.**   As our first example where Fenchel-Young and Fitzpatrick losses substantially differ, we introduce the **Fitzpatrick sparseMAP** loss, which is the Fitzpatrick counterpart of the sparseMAP loss [24].

---

**Proposition 5** *Fitzpatrick sparseMAP loss*

*When $\Omega(y') = \frac{1}{2}\|y'\|_2^2 + \iota_{\mathcal{C}}(y')$, where $\mathcal{C}$ is a nonempty closed convex set, we have for all $y \in \mathcal{C}$ and $\theta \in \mathbb{R}^k$*

$$L_{F[\partial\Omega]}(y,\theta) = 2\Omega^*\left((y+\theta)/2\right) - \langle y,\theta\rangle = \langle y^\star - y, \theta - y^\star\rangle$$

*where we used $y^\star$ as a shorthand for*

$$y^\star_{F[\partial\Omega]}(y,\theta) = \nabla\Omega^*((y+\theta)/2) = P_{\mathcal{C}}((y+\theta)/2).$$

---

A proof is given in Appendix B.6. As a special case, when $\mathcal{C} = \triangle^k$, we call the obtained loss the **Fitzpatrick sparsemax loss**, as it is the counterpart of the sparsemax loss [21]. Like the sparseMAP and sparsemax losses, these new losses rely on the Euclidean projection as a core building block. The Euclidean projection onto the probability simplex $\triangle^k$ can be computed exactly in $O(k)$ expected time and $O(k \log k)$ worst-case time [9, 22, 14, 12].

**Fitzpatrick logistic loss.** We now derive the Fitzpatrick counterpart of the logistic loss. Before stating the next proposition, we recall the definition of the Lambert function [13] $W : \mathbb{R}_+ \to \mathbb{R}_+$. The function $W$ is the inverse of the function $f : \mathbb{R}_+ \to \mathbb{R}_+$ where $f(w) = w \exp(w)$ for all $w \in \mathbb{R}_+$.

---

**Proposition 6** *Fitzpatrick logistic loss*

*When $\Omega(y') = \langle y', \log y' \rangle + \iota_{\triangle^k}(y')$, we have for all $y \in \triangle^k$ and $\theta \in \mathbb{R}^k$*

$$L_{F[\partial\Omega]}(y, \theta) = \langle y^\star - y, \theta - \log y^\star - 1 \rangle$$

*where we used $y^\star$ as a shorthand for $y^\star_{F[\partial\Omega]}(y, \theta)$ defined by*

$$y^\star_{F[\partial\Omega]}(y, \theta)_i = \begin{cases} \mathrm{e}^{-\lambda^\star} \mathrm{e}^{\theta_i}, & \text{if } y_i = 0, \\ \frac{y_i}{W(y_i \mathrm{e}^{\lambda^\star - \theta_i})}, & \text{if } y_i > 0. \end{cases}$$

---

A proof and the value of $\lambda^\star = \lambda^\star(y, \theta) \in \mathbb{R}$ are given in Appendix B.7. To obtain $\lambda^\star(y, \theta)$, we need to solve a one-dimensional root equation, which can be done using, for instance, a bisection.

## 3.3 Relation with Fenchel-Young losses

On first sight, Fitzpatrick losses and Fenchel-Young losses appear quite different. In the next proposition, we show that the Fitzpatrick loss generated by $\Omega$ is in fact equal to the Fenchel-Young loss generated by the modified, target-dependent function

$$\Omega_y(y') := \Omega(y') + D_\Omega(y, y'),$$

where $D_\Omega$ is the generalized Bregman divergence defined in (1). In particular, Lemma 1 in the appendix shows that if $\Omega = \Psi + \iota_\mathcal{C}$, where $\mathcal{C}$ is a nonempty closed convex set, then $\Omega_y(y') = \Psi_y(y') + \iota_\mathcal{C}(y')$, where $\Psi_y(y') := \Psi(y') + D_\Psi(y, y')$.

---

**Proposition 7** *Characterization of $F[\partial\Omega]$, $L_{F[\partial\Omega]}$ and $y^\star_{F[\partial\Omega]}$ using $\Omega_y$*

*Let $\Omega : \mathbb{R}^k \to \overline{\mathbb{R}}$ be a proper convex l.s.c. function. Then, for all $y \in \mathrm{dom}\,\Omega$ and all $\theta \in \mathbb{R}^k$,*

$$F[\partial\Omega](y, \theta) = \Omega_y(y) + \Omega_y^*(\theta)$$
$$L_{F[\partial\Omega]}(y, \theta) = L_{\Omega_y \oplus \Omega_y^*}(y, \theta)$$
$$y^\star_{F[\partial\Omega]}(y, \theta) = \widehat{y}_{\Omega_y}(\theta).$$

---

This characterization of the Fitzpatrick function $F[\partial\Omega]$ is also new to our knowledge. A proof is given in Appendix B.8. Proposition 7 is very useful, as it means that Fitzpatrick losses inherit from all the known properties of Fenchel-Young losses, analyzed in prior works [8, 6]. In particular, Fenchel-Young losses are smooth (i.e., with Lipschitz gradients) when $\Omega$ is strongly convex. We therefore immediately obtain that Fitzpatrick losses are smooth in their second argument $\theta$ if $\Omega$ is strongly convex and $D_\Omega$ is convex in its second argument, which is the case when $\Omega(y') = \frac{1}{2}\|y'\|_2^2$ and $\Omega(y') = \langle y', \log y' \rangle$. Therefore, the Fitzpatrick sparsemax and logistic losses are smooth. However in the general case, this does not hold, as $D_\Omega$ is usually not convex in its second argument. Proposition 7 also provides a mean to compute Fitzpatrick losses and their gradient. Finally, it suggests a very natural geometric interpretation of Fitzpatrick losses, as presented in Figure 2.

## 3.4 Relation with generalized Bregman divergences

As we stated before, the generalized Bregman divergence $D_\Omega(y, y')$ in (1) is a primal-primal divergence, as both $y$ and $y'$ belong to the same primal space. In contrast, Fenchel-Young losses

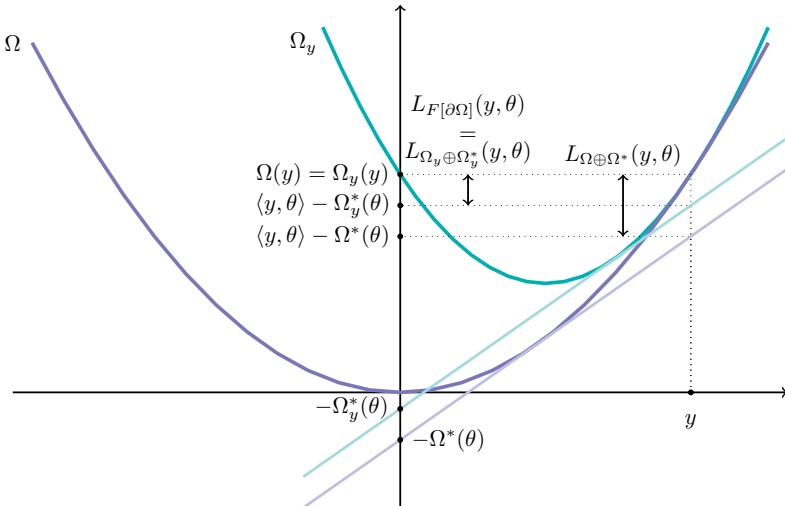

Figure 2: **Geometric interpretation**, with $\Omega(y') = \frac{1}{2}\|y'\|_2^2$. The Fenchel-Young loss $L_{\Omega \oplus \Omega^*}(y, \theta)$ is the gap (depicted with a double-headed arrow) between $\Omega(y)$ and $\langle y, \theta \rangle - \Omega^*(\theta)$, the value at $y$ of the tangent with slope $\theta$ and intercept $-\Omega^*(\theta)$. As per Proposition 7, the Fitzpatrick loss $L_{F[\partial \Omega]}(y, \theta)$ is equal to $L_{\Omega_y \oplus \Omega_y^*}(y, \theta)$ and is therefore equal to the gap between $\Omega_y(y) = \Omega(y)$ and $\langle y, \theta \rangle - \Omega_y^*(\theta)$, the value at $y$ of the tangent with slope $\theta$ and intercept $-\Omega_y^*(\theta)$. Since $\Omega_y(y') = \Omega(y') + D_\Omega(y, y')$, we have that $\Omega_y(y') \geq \Omega(y')$, with equality when $y = y'$. We therefore have $\Omega_y^*(\theta) \leq \Omega^*(\theta)$, implying that the Fitzpatrick loss is a lower bound of the Fenchel-Young loss.

$L_{\Omega \oplus \Omega^*}(y, \theta)$ are primal-dual, since $y$ belongs to the primal space and $\theta$ belongs to the dual space. Both can however be related for any proper l.s.c. convex function $\Omega$, since, for any $y'$ such that $\partial\Omega(y') \neq \emptyset$, we have

$$\inf_{\theta' \in \partial\Omega(y')} L_{\Omega \oplus \Omega^*}(y, \theta') = \inf_{\theta' \in \partial\Omega(y')} \Omega(y) + \Omega^*(\theta') - \langle y, \theta' \rangle$$

$$= \Omega(y) + \inf_{\theta' \in \partial\Omega(y')} \Omega^*(\theta') - \langle y, \theta' \rangle$$

$$= \Omega(y) - \sup_{\theta' \in \partial\Omega(y')} -\Omega^*(\theta') + \langle y, \theta' \rangle$$

$$= \Omega(y) - \Omega(y') - \sup_{\theta' \in \partial\Omega(y')} \langle y - y', \theta' \rangle$$

$$= D_\Omega(y, y')$$

where, in the penultimate line, we have used that $\Omega^*(\theta') = \langle y', \theta' \rangle - \Omega(y')$, as $\theta' \in \partial\Omega(y')$. This equality remains true when $\partial\Omega(y') = \emptyset$, by convention $\inf \emptyset = +\infty$ and by definition of $D_\Omega(y, y')$ in (1). This identity suggests that we can create Bregman-like primal-primal divergences by replacing $\Omega \oplus \Omega^*$ with $F[\partial\Omega]$,

$$\mathcal{D}_{F[\partial\Omega]}(y, y') \coloneqq \inf_{\theta' \in \partial\Omega(y')} L_{F[\partial\Omega]}(y, \theta') = \inf_{\theta' \in \partial\Omega(y')} F[\partial\Omega](y, \theta') - \langle y, \theta' \rangle.$$

This recovers one of the two Bregman-like divergences proposed in [10], the other one replacing the inf above by a sup. As stated in [10], $F[\partial\Omega]$ and $\Omega \oplus \Omega^*$ are **representations** of $\partial\Omega$.

In order to use a primal-primal divergence as a loss, we need to explicitly compose it with a link function, such as $\widehat{y}_\Omega(\theta) = \nabla\Omega^*(\theta)$. Unfortunately, $D_\Omega(y, \widehat{y}_\Omega(\theta))$ or $\mathcal{D}_{F[\partial\Omega]}(y, \widehat{y}_\Omega(\theta))$ are typically **non convex** functions of $\theta$, while Fenchel-Young and Fitzpatrick losses are always **convex** in $\theta$. In addition, differentiating through $\widehat{y}_\Omega(\theta)$ typically requires implicit differentiation [19, 5], while Fenchel-Young and Fitzpatrick losses enjoy easy-to-compute gradients, thanks to Danskin's theorem.

### 3.5 Lower bound on Fitzpatrick losses

If $\Omega = \Psi + \iota_\mathcal{C}$, where $\Psi : \mathbb{R}^k \to \overline{\mathbb{R}}$ is a proper convex Legendre-type function and $\mathcal{C} \subseteq \operatorname{dom} \Psi$ is a nonempty closed convex set, then it was shown in [8, Proposition 3] that Fenchel-Young losses

satisfy the lower bound

$$D_\Psi\big(y, \widehat{y}_\Omega(\theta)\big) \leq L_{\Omega \oplus \Omega^*}(y, \theta),$$

with equality if $\mathcal{C} = \operatorname{dom}\Psi$. We now show that a similar result holds for Fitzpatrick losses. Similarly as before, we define $\Psi_y(y') \coloneqq \Psi(y') + D_\Psi(y, y')$.

---

**Proposition 8** *Lower bound on Fitzpatrick losses*

*Let $\Omega = \Psi + \iota_\mathcal{C}$, where $\Psi : \mathbb{R}^k \to \overline{\mathbb{R}}$ is a proper convex Legendre-type function, as defined in [8, Definition 3], and $\mathcal{C} \subseteq \operatorname{dom}\Psi$ is a nonempty closed convex set. We remind that $\Omega_y(y') \coloneqq \Omega(y') + D_\Omega(y, y')$. Let us assume that $\Omega_y^*$ is differentiable. Then,*

$$D_{\Psi_y}(y, y^\star) = \langle y - y^\star, \nabla^2 \Psi(y^\star)(y - y^\star) \rangle \leq L_{F[\partial\Omega]}(y, \theta),$$

*with equality if $\operatorname{dom}\Psi = \mathcal{C}$, where we used $y^\star$ as a shorthand for $\nabla\Omega_y^*(\theta)$.*

---

A proof is given in Appendix B.9. If $\Psi_y$ is $\mu$-strongly convex, we obtain $\frac{\mu}{2}\|y - y^\star\|_2^2 \leq D_{\Psi_y}(y, y^\star)$.

## 4  Experiments

**Experimental setup.**  We follow exactly the same experimental setup as in [7, 8]. We consider a dataset of $n$ pairs $(x_i, y_i)$ of feature vectors $x_i \in \mathbb{R}^d$ and label proportions $y_i \in \triangle^k$, where $d$ is the number of features and $k$ is the number of classes. At inference time, given an unknown input vector $x \in \mathbb{R}^d$, our goal is to estimate a vector of label proportions $\widehat{y} \in \triangle^k$. A model is specified by a matrix $W \in \mathbb{R}^{k \times d}$ and a convex l.s.c. function $\Omega : \mathbb{R}^k \to \overline{\mathbb{R}}$. Predictions are then produced by the generalized linear model $x \mapsto \widehat{y}_\Omega(Wx)$. At training time, we estimate the matrix $W \in \mathbb{R}^{k \times d}$ by minimizing the convex objective

$$R_{L,\lambda}(W) \coloneqq \sum_{i=1}^n L(y_i, Wx_i) + \frac{\lambda}{2}\|W\|_2^2, \tag{5}$$

where $L \in \big\{L_{\Omega \oplus \Omega^*}, L_{F[\partial\Omega]}\big\}$. We focus on the (Fitzpatrick) sparsemax and the (Fitzpatrick) logistic losses. We optimize (5) using the L-BFGS algorithm [20]. The gradient of the Fenchel-Young loss is given in (2), while the gradient of the Fitzpatrick loss is given in Proposition 1, Item 4. Experiments were conducted on a Intel Xeon E5-2667 clocked at 3.30GHz with 192 GB of RAM running on Linux. Our implementation relies on the SciPy [29] and scikit-learn [25] libraries.

We ran experiments on 11 standard multi-label benchmark datasets[1] (see Table 2 in Appendix A for statistics on the datasets). For all datasets, we removed samples with no label, normalized samples to have zero mean unit variance, and normalized labels to lie in the probability simplex. In (5), we chose the hyperparameter $\lambda \in \{10^{-4}, 10^{-3}, \dots, 10^4\}$ against the validation set. We report test set mean squared error (also known as Brier score in a probabilistic forecasting context) in Table 1.

**Results.**  We found that the logistic loss and the Fitzpatrick logistic loss are comparable on most datasets, with the logistic loss significantly winning on 2 datasets and the Fitzpatrick logistic loss significantly winning on 2 datasets, out of 11. Since the Fitzpatrick logistic loss is slightly more computationally demanding, requiring to solve a root equation while the logistic loss does not, we believe that the logistic loss remains the best choice when we wish to use the softargmax as link function $\widehat{y}_\Omega$.

Similarly, we found that the sparsemax loss and the Fitzpatrick sparsemax loss are comparable on most datasets, with the sparsemax loss significantly winning on only 1 dataset out of 11 and the Fitzpatrick loss significantly winning on 2 datasets out of 11. Since the two losses both use the Euclidean projection onto the simplex $P_{\triangle^k}$ as their link function $\widehat{y}_\Omega$, we conclude that the Fitzpatrick sparsemax loss is a serious contender to the sparsemax loss, especially when predicting sparse label proportions is important.

---

[1]The datasets can be downloaded from `http://mulan.sourceforge.net/datasets-mlc.html` and `https://www.csie.ntu.edu.tw/~cjlin/libsvmtools/datasets/`.

| Dataset | Sparsemax | Fitzpatrick-sparsemax | Logistic | Fitzpatrick-logistic |
|---|---|---|---|---|
| Birds | 0.531 | **0.513** | 0.519 | 0.522 |
| Cal500 | 0.035 | 0.035 | 0.034 | 0.034 |
| Delicious | 0.051 | 0.052 | 0.056 | 0.055 |
| Ecthr A | 0.514 | 0.514 | 0.431 | **0.423** |
| Emotions | 0.317 | 0.318 | 0.327 | **0.320** |
| Flags | 0.186 | 0.188 | 0.184 | 0.187 |
| Mediamill | **0.191** | 0.203 | **0.207** | 0.220 |
| Scene | 0.363 | **0.355** | **0.344** | 0.368 |
| Tmc | 0.151 | 0.152 | 0.161 | 0.160 |
| Unfair | 0.149 | 0.148 | 0.157 | 0.158 |
| Yeast | 0.186 | 0.187 | 0.183 | 0.185 |

Table 1: Test performance comparison between the sparsemax loss, the logistic loss and their Fitzpatrick counterparts on the task of label proportion estimation, with regularization parameter $\lambda$ in (5) tuned against the validation set. For each dataset, label proportion errors are measured using the mean squared error (MSE). We use bold if the error is at least 0.005 lower than its counterpart.

## 5   Conclusion

We proposed to leverage the Fitzpatrick function, a theoretical tool from monotone operator theory, to build a new family of primal-dual separately convex loss functions for machine learning. We reinterpreted Fitzpatrick losses as lower bounds of Fenchel-Young losses that maintains the same link function. Our paper therefore challenges the idea that there can only be one loss function, convex in each argument, associated with a certain link function. For instance, we created the Fitzpatrick logistic and sparsemax losses, that are associated with the soft argmax and sparse argmax links, traditionally associated with the logistic and sparsemax losses, respectively. We believe that even more loss functions with the same link can be created, which calls for a systematic study of their properties and respective benefits. In future work, we intend to study calibration guarantees for Fitzpatrick losses, to test new link functions from maximal monotone operator theory and to implement more efficient training for the Fitzpatrick logistic loss.

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

## A Datasets statistics

| Dataset | Type | Train | Dev | Test | Features | Classes | Avg.labels |
|---|---|---|---|---|---|---|---|
| Birds | Audio | 134 | 45 | 172 | 260 | 19 | 2 |
| Cal500 | Music | 376 | 126 | 101 | 68 | 174 | 26 |
| Delicious | Text | 9682 | 3228 | 3181 | 500 | 983 | 19 |
| Ecthr A | Text | 6683 | 228 | 847 | 92401 | 10 | 1 |
| Emotions | Music | 293 | 98 | 202 | 72 | 6 | 2 |
| Flags | Images | 96 | 33 | 65 | 19 | 7 | 3 |
| Mediamill | Video | 22353 | 7451 | 12373 | 120 | 101 | 5 |
| Scene | Images | 908 | 303 | 1196 | 294 | 6 | 1 |
| Tmc | Text | 16139 | 5380 | 7077 | 48099 | 896 | 6 |
| Unfair | Text | 645 | 215 | 172 | 6290 | 8 | 1 |
| Yeast | Micro-array | 1125 | 375 | 917 | 103 | 14 | 4 |

Table 2: Datasets statistics

## B Proofs

### B.1 Lemmas

**Lemma 1** *Generalized Bregman divergence for constrained $\Omega$*

*Let $\Omega = \Psi + \iota_{\mathcal{C}}$, where $\Psi : \mathbb{R}^k \to \overline{\mathbb{R}}$ is proper convex l.s.c. and $\mathcal{C} \subseteq \operatorname{dom} \Psi$ is a nonempty closed convex set such that $\operatorname{ri}\mathcal{C} \cap \operatorname{ri} \operatorname{dom} \Psi \neq \emptyset$, where $\operatorname{ri}\mathcal{C}$ is the relative interior of $\mathcal{C}$. Then, for all $y, y' \in \mathbb{R}^k$*

$$D_\Omega(y, y') = D_\Psi(y, y') + D_{\iota_{\mathcal{C}}}(y, y').$$

**Proof.**

As $\mathcal{C}, \operatorname{dom} \Psi \subset \mathbb{R}^k$ and $\operatorname{ri}\mathcal{C} \cap \operatorname{ri} \operatorname{dom} \Psi \neq \emptyset$, we can apply [3, Proposition 6.19] and [3, Theorem 16.47] to write $\partial\Omega(y') = \partial\Psi(y') + \partial\iota_{\mathcal{C}}(y')$.

Thus, we have

$$D_\Omega(y, y') := \Omega(y) - \Omega(y') - \sup_{\theta' \in \partial\Omega(y')} \langle y - y', \theta' \rangle$$

$$= \Psi(y) + \iota_{\mathcal{C}}(y) - \Psi(y') - \iota_{\mathcal{C}}(y') - \sup_{\theta' \in \partial\Psi(y') + \partial\iota_{\mathcal{C}}(y')} \langle y - y', \theta' \rangle$$

$$= \Psi(y) - \Psi(y') - \sup_{\theta' \in \partial\Psi(y')} \langle y - y', \theta' \rangle + \iota_{\mathcal{C}}(y) - \iota_{\mathcal{C}}(y') - \sup_{\theta' \partial\iota_{\mathcal{C}}(y')} \langle y - y', \theta' \rangle$$

$$= D_\Psi(y, y') + D_{\iota_{\mathcal{C}}}(y, y').$$

**Lemma 2** *Generalized Bregman divergence of indicator function of some nonempty closed convex set set $\mathcal{C} \subset \mathbb{R}^k$*

$$D_{\iota_{\mathcal{C}}}(y, y') = \begin{cases} \iota_{\mathcal{C}}(y) & \text{if } y' \in \mathcal{C} \\ \infty & \text{if } y' \notin \mathcal{C} \end{cases} = \iota_{\mathcal{C}}(y) + \iota_{\mathcal{C}}(y').$$

**Proof.** Using (1) and the fact that $\partial\iota_{\mathcal{C}} = N_{\mathcal{C}}$ [3, Example 16.13], we obtain that

$$D_{\iota_{\mathcal{C}}}(y, y') := \iota_{\mathcal{C}}(y) - \iota_{\mathcal{C}}(y') - \sup_{\theta' \in N_{\mathcal{C}}(y')} \langle y - y', \theta' \rangle.$$

When $y' \in \mathcal{C}$ and $y \in \mathcal{C}$,

$$\sup_{\theta' \in N_{\mathcal{C}}(y')} \langle y - y', \theta' \rangle = \sup_{\substack{\theta' \in \mathbb{R}^k \\ \langle z - y', \theta' \rangle \leq 0 \\ \forall z \in \mathcal{C}}} \langle y - y', \theta' \rangle = 0.$$

When $y' \in \mathcal{C}$ and $y \notin \mathcal{C}$, $D_{\iota_{\mathcal{C}}}(y, y') = +\infty$, as $+\infty + (-\infty) = +\infty$ in the definition of the Bregman divergence. Therefore, when $y' \in \mathcal{C}$, we get that $D_{\iota_{\mathcal{C}}}(y, y') = \iota_{\mathcal{C}}(y)$.

When $y' \notin \mathcal{C}$, $N_{\mathcal{C}}(y') = \emptyset$. Again, in the definition of the Bregman divergence, $+\infty + (-\infty) = +\infty$ and we use the convention $\sup_{\emptyset} = -\infty$.

---

**Lemma 3** *Bregman divergence of $\Psi_y$*

*Let $\Psi : \mathbb{R}^k \to \overline{\mathbb{R}}$ be proper, convex and twice differentiable on the interior of its domain. For all $y \in \mathbb{R}^k$, let $\Psi_y := \Psi + D_{\Psi}(y, \cdot)$. Then, for all $y, y', y'' \in \operatorname{int} \operatorname{dom} \Psi$,*

$$D_{\Psi_y}(y'', y') = D_{\Psi}(y, y'') - D_{\Psi}(y, y') + D_{\Psi}(y'', y') + \langle y'' - y', \nabla^2 \Psi(y')(y - y') \rangle$$
$$= \langle y'' - y, \nabla \Psi(y'') \rangle - \langle y'' - y, \nabla \Psi(y') \rangle + \langle y'' - y', \nabla^2 \Psi(y')(y - y') \rangle$$

*and, in particular, for all $y, y' \in \operatorname{int} \operatorname{dom} \Psi$*

$$D_{\Psi_y}(y, y') = \langle y - y', \nabla^2 \Psi(y')(y - y') \rangle.$$

---

**Proof.** For all $y, y' \in \operatorname{int} \operatorname{dom} \Psi$, as $\Psi$ is convex and differentiable on $\operatorname{int} \operatorname{dom} \Psi$, we have that

$$\begin{aligned}
\Psi_y(y') &= \Psi(y') + D_{\Psi}(y, y') \\
&= \Psi(y') + \Psi(y) - \Psi(y') - \langle y - y', \nabla \Psi(y') \rangle \\
&= \Psi(y) - \langle y - y', \nabla \Psi(y') \rangle,
\end{aligned}$$

and therefore, as $\Psi$ is twice differentiable on $\operatorname{int} \operatorname{dom} \Psi$, we get that

$$\nabla \Psi_y(y') = \nabla^2 \Psi(y')(y' - y) + \nabla \Psi(y').$$

Therefore, for all $y, y', y'' \in \operatorname{dom} \Psi$,

$$\begin{aligned}
D_{\Psi_y}(y'', y') &= \Psi_y(y'') - \Psi_y(y') - \langle y'' - y', \nabla \Psi_y(y') \rangle \\
&= \Psi(y'') + D_{\Psi}(y, y'') - \Psi(y') - D_{\Psi}(y, y') - \langle y'' - y', \nabla^2 \Psi(y')(y' - y) \rangle \\
&\quad - \langle y'' - y', \nabla \Psi(y') \rangle \\
&= D_{\Psi}(y, y'') - D_{\Psi}(y, y') + D_{\Psi}(y'', y') + \langle y'' - y', \nabla^2 \Psi(y')(y - y') \rangle \\
&= \langle y'' - y, \nabla \Psi(y'') \rangle - \langle y'' - y, \nabla \Psi(y') \rangle + \langle y'' - y', \nabla^2 \Psi(y')(y - y') \rangle
\end{aligned}$$

and in particular using the just last established equality with the triplet $(y, y, y')$, we have for all $y, y' \in \operatorname{dom} \Psi$,

$$\begin{aligned}
D_{\Psi_y}(y, y') &= \langle y - y, \nabla \Psi(y) \rangle - \langle y - y, \nabla \Psi(y') \rangle + \langle y - y', \nabla^2 \Psi(y')(y - y') \rangle \\
&= \langle y - y', \nabla^2 \Psi(y')(y - y') \rangle.
\end{aligned}$$

---

**Lemma 4** *Generalized Bregman divergence of negentropy*

*Let $\alpha \in \mathbb{R}$. Let $\Psi(y') := \sum_{i=1}^{k} y'_i \log y'_i - \alpha \sum_{i=1}^{k} y'_i$ be defined for $y' \in \mathbb{R}^k_+$. Then, for $y, y' \in \mathbb{R}^k_+$,*

$$D_{\Psi}(y, y') = \sum_{i=1}^{k} y_i \log \frac{y_i}{y'_i} - \sum_{i=1}^{k} (y_i - y'_i) + \iota_{\mathbb{R}^k_{++}}(y'),$$

*which does not depend on $\alpha$.*

---

**Proof.** First, if $y' \in \mathbb{R}^k_{++}$, $\Psi$ is differentiable at $y'$ and $\nabla \Psi(y')_i = \log y_i + 1 - \alpha$. Thus, $\partial \Psi(y') = \{\nabla \Psi(y')\}$ and

$$\begin{aligned}
D_{\Psi}(y, y') &= \Psi(y) - \Psi(y') - \sup_{\theta' \in \partial \Psi(y')} \langle y - y', \theta' \rangle, \\
&= \Psi(y) - \Psi(y') - \langle y - y', \nabla \Psi(y') \rangle, \\
&= \sum_{i=1}^{k} y_i \log \frac{y_i}{y'_i} - \sum_{i=1}^{k} (y_i - y'_i).
\end{aligned}$$

Second, if we prove that $\partial \Psi(y') = \emptyset$ when there exists a component $i$ such that $y'_i = 0$, we can conclude the proof, as $\sup_\emptyset = -\infty$ by convention. Indeed, Let us assume that $y'_i = 0$. Suppose that $\theta' \in \partial \Psi(y')$. Then, by definition of subgradients,

$$\langle y'' - y', \theta' \rangle + \Psi(y') \leq \Psi(y''), \ \forall y'' \in \mathbb{R}^k_{++}.$$

We consider $\varepsilon > 0$ and choose $y'' = y' + \varepsilon e_i$, where $e_i$ is the $i$-th canonical base vector. Thus, we obtain

$$\varepsilon \theta_i \leq \Psi(y' + \varepsilon e_i) - \Psi(y'),$$

$$= \sum_{j=1}^k y'_j \log y'_j + \varepsilon \log \varepsilon - \alpha \sum_{j=1}^k y'_j - \alpha \varepsilon - \Big( \sum_{j=1}^k y'_j \log y'_j - \alpha \sum_{j=1}^k y'_j \Big),$$

$$= \varepsilon \log \varepsilon - \alpha \varepsilon,$$

as $y_i = 0$ and $0 \log 0 = 0$ by convention. By noticing that $\lim_{\varepsilon \to 0^+} (\varepsilon \log \varepsilon - \alpha \varepsilon)/\varepsilon = -\infty$, we get a contradiction, which concludes the proof.

---

**Lemma 5** *Subdifferential inclusion for* sup *function*

*Let $Y \subset \mathbb{R}^k$ be some set. Let $\{f_y\}_{y \in Y}$ be a family of finite functions $f_y : \mathbb{R}^k \to \mathbb{R}$. We define the function $g : \mathbb{R}^k \to \overline{\mathbb{R}}$ by $g(\theta) = \sup_{y \in Y} f_y(\theta)$, for any $\theta \in \mathbb{R}^k$. Then, we have*

$$\partial f_{\tilde{y}}(\theta) \subset \partial g(\theta),$$

*for all $\theta \in \mathbb{R}^k$ and for all $\tilde{y} \in \operatorname{argmax}_{y \in Y} f_y(\theta)$.*

---

**Proof.** Left as an exercise.

---

**Lemma 6** *Value and gradient of $\Psi^*_y$*

*Let $\Psi : \mathbb{R}^k \to \overline{\mathbb{R}}$ be a proper strictly convex function. Let us assume that $D_\Psi(y, y')$ is convex w.r.t. $y'$, for all $y \in \operatorname{dom} \Psi$. We remind that $\Psi_y(y') = \Psi(y') + D_\Psi(y, y')$. Then, for all $\theta \in \mathbb{R}^k$ and for all $\tilde{y} \in \operatorname{argmax}_{y' \in \operatorname{dom} \Psi} \langle y', \theta \rangle + \langle y - y', \nabla \Psi(y') \rangle$,*

$$\Psi^*_y(\theta) = \langle \tilde{y}, \theta \rangle - \Psi(y) + \langle y - \tilde{y}, \nabla \Psi(\tilde{y}) \rangle$$

$$\nabla \Psi^*_y(\theta) = \tilde{y}.$$

*Furthermore, if the function $\Psi$ is twice differentiable, we have*

$$\operatorname*{argmax}_{y' \in \operatorname{dom} \Psi} \langle y', \theta \rangle + \langle y - y', \nabla \Psi(y') \rangle \subset \big\{ y' \in \operatorname{dom} \Psi \ \big| \ \nabla^2 \Psi(y')(y' - y) = \theta - \nabla \Psi(y') \big\}.$$

---

**Proof.** As $\Psi \leq \Psi_y$, we have $\operatorname{dom} \Psi_y \subset \operatorname{dom} \Psi$. Thus, we get

$$\Psi^*_y(\theta) = \sup_{y' \in \operatorname{dom} \Psi} \langle y', \theta \rangle - \Psi_y(y')$$

$$= \sup_{y' \in \operatorname{dom} \Psi} \langle y', \theta \rangle - \big( \Psi(y') + \Psi(y) - \Psi(y') - \langle y - y', \nabla \Psi(y') \rangle \big)$$

$$= \sup_{y' \in \operatorname{dom} \Psi} \langle y', \theta \rangle - \Psi(y) + \langle y - y', \nabla \Psi(y') \rangle.$$

As $\Psi$ is strictly convex and $D_\Psi(y, y')$ is convex w.r.t. $y'$, we have that $\Psi_y$ is strictly convex. Thus, according to [27, Theorem 11.13], $\Psi^*_y$ is differentiable. Thus,

$$\{ \nabla \Psi^*_y(\theta) \} = \partial \Psi^*_y(\theta)$$

$$\supset \operatorname*{argmax}_{y' \in \operatorname{dom} \Psi} \langle y', \theta \rangle - \Psi(y) + \langle y - y', \nabla \Psi(y') \rangle$$

$$= \operatorname*{argmax}_{y' \in \operatorname{dom} \Psi} \langle y', \theta \rangle + \langle y - y', \nabla \Psi(y') \rangle,$$

using Lemma 5 for the inclusion.

In the case that $\Psi$ is twice differentiable, setting the gradient of the inner function to zero concludes the proof.

> **Lemma 7** *Gradient of $\Psi_y^*$, squared norm case*
>
> Let $\Psi(y') := \frac{1}{2}\|y'\|_2^2$ and $\Psi_y$ defined as in Lemma 6. Then, the gradient of $\Psi_y^*$ is given by
>
> $$\nabla\Psi_y^*(\theta) = \frac{y+\theta}{2}.$$

**Proof.** Let us notice that

$$\sup_{y'\in\text{dom }\Psi} \langle y',\theta\rangle + \langle y-y',\nabla\Psi(y')\rangle = \sup_{y'\in\text{dom }\Psi} \langle y',\theta+y'\rangle - \|y'\|^2,$$

and that $\Psi$ is strictly convex function. By strong convexity of $y' \mapsto \|y'\|^2$, we deduce that $\text{argmax}_{y'\in\text{dom }\Psi}\langle y',\theta\rangle + \langle y-y',\nabla\Psi(y')\rangle \neq \emptyset$. Furthermore, as $\Psi$ is twice differentiable and $\nabla^2\Psi(y') = I$ and $\text{dom }\Psi = \mathbb{R}^k$, we have that $\{y' \in \text{dom }\Psi \mid \nabla^2\Psi(y')(y'-y) = \theta - \nabla\Psi(y')\}$ is a singleton. Thus, we use Lemma 6 with $\nabla\Psi(y') = y'$ and $\nabla^2\Psi(y') = I$, we obtain that $\nabla\Psi_y^*(\theta)$ is the solution w.r.t. $y'$ of the equation $y' - y = \theta - y'$. Rearranging the terms concludes the proof.

Before stating the next lemma, we recall the definition of the Lambert function [13] $W : \mathbb{R}_+ \to \mathbb{R}_+$. The function $W$ is the inverse of the function $f : \mathbb{R}_+ \to \mathbb{R}_+$ where $f(w) = w \exp(w)$ for all $w \in \mathbb{R}_+$.

---

**Lemma 8** *Gradient of $\Psi_y^*$, negentropy case*

Let $\Psi(y') := \sum_{i=1}^k y_i' \log y_i' - \alpha \sum_{i=1}^k y_i'$ be defined for $y' \in \mathbb{R}_+^k$. Then,

$$\nabla \Psi_y^*(\theta)_i = \left\{ \begin{array}{ll} \mathrm{e}^{\theta_i - 2 + \alpha}, & \text{if } y_i = 0 \\ \frac{y_i}{W(y_i \mathrm{e}^{-(\theta_i - 2 + \alpha)})}, & \text{if } y_i > 0. \end{array} \right.$$

---

**Proof.** Following the same arguments as in the proof of Lemma 7, we use Lemma 6. We know that $\tilde{y}$ is the solution of $\nabla^2 \Psi(\tilde{y})(\tilde{y} - y) = \theta - \nabla \Psi(\tilde{y})$. Using $\nabla \Psi(\tilde{y}) = \log \tilde{y} + 1 - \alpha$ and $\nabla^2 \Psi(\tilde{y}) = 1/\tilde{y}$ (where logarithm and division are performed element-wise), we obtain for all $i \in \{1, \ldots, k\}$

$$(\tilde{y}_i - y_i)/\tilde{y}_i = \theta_i - \log \tilde{y}_i - 1 + \alpha \iff 1 - y_i/\tilde{y}_i = \theta_i - \log \tilde{y}_i - 1 + \alpha.$$

When $y_i = 0$, we immediately have $\tilde{y}_i = \exp(\theta_i - 2 + \alpha)$. When $y_i > 0$, after rearranging, we obtain

$$\frac{y_i}{\tilde{y}_i} \exp\left(\frac{y_i}{\tilde{y}_i}\right) = y_i \exp(-(\theta_i - 2 + \alpha)) \iff \frac{y_i}{\tilde{y}_i} = W(y_i \exp(-(\theta_i - 2 + \alpha))),$$

hence the result.

---

**Lemma 9** *Gradient of $\Omega_y^*$*

Let $\Psi : \mathbb{R}^k \to \overline{\mathbb{R}}$ be a proper strictly convex l.s.c. function and assume that, for all $y \in \mathbb{R}^k$, the function $D_\Psi(y, \cdot)$ is convex. Let $\Omega_y = \Psi_y + \iota_{\mathcal{C}}$ with $\Psi_y$ defined as in Lemma 6 and assume that $\mathcal{C}$ is a nonempty compact convex set included in $\mathrm{dom}\,\Psi$. Then $\nabla \Omega_y^*(\theta)$ is the unique element of the set

$$\underset{y' \in \mathcal{C}}{\mathrm{argmax}} \ \langle y', \theta \rangle + \langle y - y', \nabla \Psi(y') \rangle.$$

---

**Proof.** The result follows from Danskin's theorem [4, Proposition B.25].

---

**Lemma 10** *Dual of simplex-constrained conjugate*

Let $\Psi : \mathbb{R}^k \to \overline{\mathbb{R}}$ be a proper strictly convex function. Then,

$$(\Psi + \iota_{\triangle^k})^*(\theta) = \min_{\tau \in \mathbb{R}} \tau + (\Psi + \iota_{\mathbb{R}_+^k})^*(\theta - \tau \mathbf{1}).$$

and for any $\tau^\star \in \mathrm{argmin}_{\tau \in \mathbb{R}} \tau + (\Psi + \iota_{\mathbb{R}_+^k})^*(\theta - \tau \mathbf{1})$, we get

$$\nabla(\Psi + \iota_{\triangle^k})^*(\theta) = \nabla(\Psi + \iota_{\mathbb{R}_+^k})^*(\theta - \tau^\star \mathbf{1}).$$

---

**Proof.** We have by the definition of the Fenchel conjugate that $(\Psi + \iota_{\triangle^k})^*(\theta) = \max_{y' \in \triangle^k} (\langle y', \theta \rangle - \Psi(y'))$. As the unit simplex $\triangle^k$ is a compact set and as $(\Psi + \iota_{\triangle^k})^*$ is differentiable (by [27, Theorem 11.13] given that $\Psi$ is strictly convex and $\triangle^k$ is convex ), we apply Danskin's theorem [4, Proposition B.25] and get $\{\nabla(\Psi + \iota_{\triangle^k})^*(\theta)\} = \mathrm{argmax}_{y' \in \triangle^k} (\langle y', \theta \rangle - \Psi(y'))$.

Furthermore, we rewrite $(\Psi + \iota_{\triangle^k})^*$ in the following dual minimization problem

$$\begin{aligned} (\Psi + \iota_{\triangle^k})^*(\theta) &= \max_{y' \in \triangle^k} \langle y', \theta \rangle - \Psi(y') \\ &= \max_{y' \in \mathbb{R}_+^k} \min_{\tau \in \mathbb{R}} \langle y', \theta \rangle - \Psi(y') - \tau(\langle y', \mathbf{1} \rangle - 1) \\ &= \min_{\tau \in \mathbb{R}} \tau + \max_{y' \in \mathbb{R}_+^k} \langle y', \theta - \tau \mathbf{1} \rangle - \Psi(y') \\ &= \min_{\tau \in \mathbb{R}} \tau + (\Psi + \iota_{\mathbb{R}_+^k})^*(\theta - \tau \mathbf{1}). \end{aligned}$$

Again, as $\Psi$ is strictly convex, $(\Psi + \iota_{\mathbb{R}_+^k})^*$ is differentiable [27, Theorem 11.13]. After some elementary calculus on subdifferentials, we conclude by applying primal and dual forms of Fermat's rule [27, Theorem 11.39, Item (d)]: for any $\tau^\star \in \mathrm{argmin}_{\tau \in \mathbb{R}} \tau + (\Psi + \iota_{\mathbb{R}_+^k})^*(\theta - \tau \mathbf{1})$, we have $\nabla(\Psi + \iota_{\mathbb{R}_+^k})^*(\theta - \tau^\star \mathbf{1}) \in \mathrm{argmax}_{y' \in \triangle^k} (\langle y', \theta \rangle - \Psi(y'))$.

**Lemma 11** *Gradient of $\Omega_{y'}^*$, negentropy, constrained to the simplex*

*Let $\Psi(y') = \langle y', \log y' \rangle$, if $y' \in \mathbb{R}_+^k$, $+\infty$ otherwise. Then, for all $y \in \mathbb{R}_+^k$, the gradient of $\Omega_y = \Psi_y + \iota_{\triangle^k}$ (as defined in Lemma 9) is given by*

$$\nabla \Omega_y^*(\theta)_i = \begin{cases} e^{-\lambda^\star} e^{\theta_i} & \text{if } y_i = 0, \\ \dfrac{y_i}{W(y_i e^{\lambda^\star - \theta_i})} & \text{if } y_i > 0. \end{cases}$$

*where $\lambda^\star$ is the unique solution of the equation*

$$e^{-\lambda^\star} \sum_{i:y_i=0} e^{\theta_i} + \sum_{i:y_i>0} \frac{y_i}{W(y_i e^{-(\theta_i - \lambda^\star)})} = 1.$$

**Proof.** From Lemma 9 and Lemma 10, since $\operatorname{dom} \Psi_y = \mathbb{R}_+^k$, we have

$$y^\star = \nabla \Omega_y^*(\theta) = \nabla \Psi_y^*(\theta - \tau^\star \mathbf{1})$$

for any solution $\tau^\star$ of

$$\min_{\tau \in \mathbb{R}} \tau + \Psi_y^*(\theta - \tau \mathbf{1}).$$

As $\tau \mapsto \tau + \Psi_y^*(\theta - \tau \mathbf{1})$ is a convex function, Setting the gradient of the inner function to zero, we get

$$\tau^\star \in \operatorname*{argmin}_{\tau \in \mathbb{R}} \tau + \Psi_y^*(\theta - \tau \mathbf{1}) \iff \langle \nabla \Psi_y^*(\theta - \tau^\star \mathbf{1}), \mathbf{1} \rangle = 1.$$

Using Lemma 8, we obtain that $\tau^\star$ satisfies

$$e^{-\tau^\star - 2} \sum_{i:y_i=0} e^{\theta_i} + \sum_{i:y_i>0} \frac{y_i}{W(y_i e^{-(\theta_i - \tau^\star - 2)})} = 1.$$

By monotonicity in $\tau^\star$, we conclude that $\tau^\star$ exists and is unique. Using the change of variable $\tau^\star = \lambda^\star + 2$ concludes the proof.

## B.2 Proof of Proposition 1 (Properties of Fitzpatrick losses)

Apart from differentiability, the proofs follow from the study of Fitzpatrick functions found in [15, 2, 28]. We include the proofs for completeness.

**Link function and non-negativity.** We recall that

$$\begin{aligned} L_{F[\partial\Omega]}(y, \theta) &= \sup_{(y', \theta') \in \partial\Omega} \langle y' - y, \theta - \theta' \rangle \\ &= -\inf_{(y', \theta') \in \partial\Omega} \langle y' - y, \theta' - \theta \rangle. \end{aligned}$$

From the monotonicity of $\partial\Omega$, we have that if $(y, \theta) \in \partial\Omega$ and $(y', \theta') \in \partial\Omega$, then $\langle y' - y, \theta' - \theta \rangle \geq 0$. Therefore, for all $(y, \theta) \in \partial\Omega$,

$$\inf_{(y', \theta') \in \partial\Omega} \langle y' - y, \theta' - \theta \rangle = 0,$$

with the infimum being attained at $(y', \theta') = (y, \theta)$. This proves the link function.

From the maximality of $\partial\Omega$, if $(y, \theta) \notin \partial\Omega$, there exists $(y', \theta') \in \partial\Omega$ such that $\langle y' - y, \theta' - \theta \rangle < 0$. Therefore, for all $(y, \theta) \notin \partial\Omega$,

$$\inf_{(y', \theta') \in \partial\Omega} \langle y' - y, \theta' - \theta \rangle < 0.$$

This proves the non-negativity.

**Separable convexity.** We recall that
$$L_{F[\partial\Omega]}(y,\theta) = F[\partial\Omega](y,\theta) - \langle y, \theta \rangle$$
where
$$F[\partial\Omega](y,\theta) = \sup_{(y',\theta')\in\partial\Omega} \langle y-y', \theta' \rangle + \langle y', \theta \rangle = \sup_{(y',\theta')\in\partial\Omega} \langle y', \theta \rangle + \langle y, \theta' \rangle - \langle y', \theta' \rangle.$$
The function $(y,\theta) \mapsto \langle y', \theta \rangle + \langle y, \theta' \rangle - \langle y', \theta' \rangle$ is (jointly) convex in $(y,\theta)$ for all $(y',\theta')$. Since the supremum preserves convexity, $F[\partial\Omega](y,\theta)$ is (jointly) convex in $(y,\theta)$. The function $\langle y, \theta \rangle$ is convex in $y$ and $\theta$ but not (jointly) convex in $(y,\theta)$. Therefore, $L_{F[\partial\Omega]}(y,\theta)$ is separately convex in $y$ and $\theta$.

**Subgradient** We are going to prove that, for any $y, \theta \in \mathbb{R}^k$ and for any $y^\star \in y^\star_{F[\partial\Omega]}(y,\theta)$ as defined in (4), we have $y^\star - y \in \partial_\theta L_{F[\partial\Omega]}(y,\theta)$.

Indeed, by Definition 1, we have that
$$L_{F[\partial\Omega]}(y,\theta) = \sup_{y'\in\text{dom }\Omega} \left[ \langle y'-y, \theta \rangle + \sup_{\theta'\in\partial\Omega(y')} \langle y-y', \theta' \rangle \right].$$
Furthermore, $\partial f_{y'}(\theta) = \{y' - y\}$, where $f_{y'}(\theta) = \langle y', \theta \rangle + \sup_{\theta'\in\partial\Omega(y')}\langle y-y', \theta' \rangle$. Thus, by applying Lemma 5, we get
$$y^\star - y \in \partial_\theta L_{F[\partial\Omega]}(y,\theta)$$
for any $y^\star \in \text{argmax}_{y'\in\text{dom }\Omega} f_{y'}(\theta) = \text{argmax}_{y'\in\text{dom }\Omega} \left[ \langle y'-y, \theta \rangle + \sup_{\theta'\in\partial\Omega(y')}\langle y-y', \theta' \rangle \right]$,
which yields the result by definition of $y^\star_{F[\partial\Omega]}(y,\theta)$ in (4).

**Differentiability.** Since the function $\Omega$ is strictly convex and $y' \mapsto D_\Omega(y,y')$ is convex, then $\Omega_y(y') = \Omega(y') + D_\Omega(y,y')$ is strictly convex in $y'$. From the duality between strict convexity and differentiability, $\Omega^*_y(\theta)$ is differentiable in $\theta$.

**Tighter inequality.** Using
$$\partial\Omega = \{(y',\theta') \in \mathbb{R}^k \times \mathbb{R}^k \mid \Omega(y) \geq \Omega(y') + \langle y-y', \theta' \rangle \ \forall y\}$$
and
$$\Omega^*(\theta) = \sup_{y'\in\mathbb{R}^k} \langle y', \theta \rangle - \Omega(y'),$$
we get for any $(y',\theta') \in \partial\Omega$,
$$\langle y-y', \theta' \rangle + \langle y', \theta \rangle \leq \Omega(y) - \Omega(y') + \langle y', \theta \rangle$$
$$\leq \Omega(y) + \Omega^*(\theta).$$
Therefore
$$F[\partial\Omega](y,\theta) = \sup_{(y',\theta')\in\partial\Omega} \langle y-y', \theta' \rangle + \langle y', \theta \rangle \leq \Omega(y) + \Omega^*(\theta).$$

### B.3 Proof of Proposition 2 (Expression of Fitzpatrick loss when $\Omega$ is twice differentiable)

We recall that
$$F[\partial\Omega](y,\theta) = \sup_{(y',\theta')\in\partial\Omega} \langle y, \theta' \rangle + \langle y', \theta \rangle - \langle y', \theta' \rangle = \sup_{y'\in\text{dom }\Omega} \langle y', \theta \rangle + \sup_{\theta'\in\partial\Omega(y')} \langle y, \theta' \rangle - \langle y', \theta' \rangle.$$
Since $\Omega$ is differentiable, we have $\partial\Omega(y') = \{\nabla\Omega(y')\}$ and therefore $\theta' = \nabla\Omega(y')$, which gives
$$F[\partial\Omega](y,\theta) = \sup_{y'\in\text{dom }\Omega} \langle y, \nabla\Omega(y') \rangle + \langle y', \theta \rangle - \langle y', \nabla\Omega(y') \rangle.$$
Setting the gradient of the inner function w.r.t. $y'$ to zero, we get, for any $y^\star \in y^\star_{F[\partial\Omega]}(y,\theta)$ as defined in (4),
$$\nabla^2\Omega(y^\star)y + \theta - \nabla\Omega(y') - \nabla^2\Omega(y^\star)y^\star = 0.$$
Using the $y' = y^\star$ and $\theta' = \nabla\Omega(y^\star)$ in Definition 1, we then obtain
$$L_{F[\partial\Omega]}(y,\theta) = \langle y'-y, \theta-\theta' \rangle$$
$$= \langle y^\star - y, \theta - \nabla\Omega(y^\star) \rangle$$
$$= \langle y^\star - y, \nabla^2\Omega(y^\star)(y^\star - y) \rangle.$$

## B.4 Proof of Proposition 3 (squared loss)

Using Proposition 2 with $\nabla\Omega(y') = y'$ and $\nabla^2\Omega(y') = I$, we obtain

$$y + \theta - 2y' = 0 \iff y' = \frac{y + \theta}{2}.$$

We therefore obtain

$$
\begin{aligned}
L_{F[\partial\Omega]}(y, \theta) &= \left\langle \frac{y + \theta}{2} - y, \theta - \frac{y + \theta}{2} \right\rangle \\
&= \left\langle \frac{\theta - y}{2}, \frac{\theta - y}{2} \right\rangle \\
&= \frac{1}{4} \|y - \theta\|_2^2.
\end{aligned}
$$

## B.5 Proof of Proposition 4 (perceptron loss)

A proof of the Fitzpatrick function for this case was given in [2, Example 3.1]. We include a proof for completeness. Since $\Omega = \iota_{\mathcal{C}}$, we have $\mathrm{dom}\,\Omega = \mathcal{C}$. Therefore, for all $y \in \mathcal{C}$ and $\theta \in \mathbb{R}^k$,

$$
\begin{aligned}
F[\partial\Omega](y, \theta) &= \sup_{y' \in \mathrm{dom}\,\Omega} \langle y', \theta \rangle + \sup_{\theta' \in \partial\Omega(y')} \langle y - y', \theta' \rangle \\
&= \sup_{y' \in \mathcal{C}} \langle y', \theta \rangle - \left( \iota_{\mathcal{C}}(y) - \iota_{\mathcal{C}}(y') - \sup_{\theta' \in \partial\iota_{\mathcal{C}}(y')} \langle y - y', \theta' \rangle \right) \\
&= \sup_{y' \in \mathcal{C}} \langle y', \theta \rangle - D_{\iota_{\mathcal{C}}}(y, y') \\
&= \sup_{y' \in \mathcal{C}} \langle y', \theta \rangle,
\end{aligned}
$$

where in the third line we used that $\iota_{\mathcal{C}}(y) = \iota_{\mathcal{C}}(y') = 0$ and where in the last line we used Lemma 2. Therefore, for all $y \in \mathbb{R}^k$ and $\theta \in \mathbb{R}^k$,

$$F[\partial\Omega](y, \theta) = \sup_{y' \in \mathcal{C}} \langle y', \theta \rangle + \iota_{\mathcal{C}}(y) = \iota_{\mathcal{C}}(y) + \iota_{\mathcal{C}}^*(\theta).$$

## B.6 Proof of Proposition 5 (Fitzpatrick sparseMAP loss)

A proof of the Fitzpatrick function for this case was given in [2, Example 3.13]. We provide an alternative proof.

From Proposition 7, we know that for any $y \in \mathrm{dom}\,\Omega (= \mathcal{C}$ here)

$$F[\partial\Omega](y, \theta) = \Omega_y(y) + \Omega_y^*(\theta) = \Omega(y) + \Omega_y^*(\theta),$$

where

$$
\begin{aligned}
\Omega_y(y') &= \frac{1}{2} \|y'\|_2^2 + \frac{1}{2} \|y - y'\|_2^2 + \iota_{\mathcal{C}}(y') \\
&= \|y'\|_2^2 + \frac{1}{2} \|y\|_2^2 - \langle y, y' \rangle + \iota_{\mathcal{C}}(y') \\
&= 2\Omega(y') + \Omega(y) - \langle y, y' \rangle.
\end{aligned}
$$

Using conjugate calculus, we obtain

$$\Omega_y^*(\theta) = 2\Omega^* \left( \frac{y + \theta}{2} \right) - \Omega(y).$$

Therefore,

$$F[\partial\Omega](y, \theta) = 2\Omega^* \left( \frac{y + \theta}{2} \right).$$

From Proposition 7, the supremum w.r.t. $y'$ is achieved at $y^\star = \nabla\Omega^*((y + \theta)/2) = P_{\mathcal{C}}((y + \theta)/2)$. We therefore obtain

$$L_{F[\partial\Omega]}(y, \theta) = \langle y^\star - y, \theta - y^\star \rangle.$$

## B.7 Proof of Proposition 6 (Fitzpatrick logistic loss)

**Differentiability w.r.t. $\theta$ and formula of gradient.** According to Proposition 7, we have

$$L_{F[\partial\Omega]}(y,\theta) = \Omega_y(y) + \Omega_y^*(\theta) - \langle y,\theta \rangle.$$

Thus the differentiability w.r.t. $\theta$ of $L_{F[\partial\Omega]}(y,\theta)$ follows from the differentiability of $\theta \mapsto \Omega_y^*(\theta)$. Lemma 11 yields the differentiability of $\Omega_y^*(\theta)$ and a formula for its gradient $y_{F[\partial\Omega]}^\star(y,\theta) := \nabla\Omega_y^*(\theta)$

$$y_{F[\partial\Omega]}^\star(y,\theta)_i = \left\{ \begin{array}{ll} e^{-\lambda^\star}e^{\theta_i}, & \text{if } y_i = 0, \\ \frac{y_i}{W(y_i e^{\lambda^\star - \theta_i})}, & \text{if } y_i > 0. \end{array} \right.$$

It follows that $\nabla_\theta L_{F[\partial\Omega]}(y,\theta) = y_{F[\partial\Omega]}^\star(y,\theta) - y$.

**Formula of the Fitzpatrick logistic loss.** We use $y^\star$ as a shorthand for $y_{F[\partial\Omega]}^\star(y,\theta)$. As we know that $\Omega_y^*(\theta) = \langle y^\star,\theta \rangle - \Omega_y(y^\star)$, we use again Proposition 7 to get

$$\begin{aligned} L_{F[\partial\Omega]}(y,\theta) &= \Omega_y(y) + \langle y^\star,\theta \rangle - \Omega_y(y^\star) - \langle y,\theta \rangle \\ &= \Omega(y) - (\Omega(y^\star) + D_\Omega(y,y^\star)) + \langle y^\star - y,\theta \rangle \end{aligned}$$

as $\Omega_y(y') = \Omega(y) + D_\Omega(y,y')$ and in particular $\Omega_y(y) = \Omega(y)$. Furthermore, as $y^\star \in \triangle^k \cap \mathbb{R}_{++}^k$, $\Omega$ is differentiable at $y^\star$ and $D_\Omega(y,y^\star) = \Omega(y) - \Omega(y^\star) - \langle y - y^\star, \nabla\Omega(y^\star) \rangle$, where $\nabla\Omega(y^\star) = \log y^\star + 1$. Thus

$$\begin{aligned} L_{F[\partial\Omega]}(y,\theta) &= \langle y - y^\star, \nabla\Omega(y^\star) \rangle + \langle y^\star - y,\theta \rangle \\ &= \langle y^\star - y, \theta - \log y^\star - 1 \rangle. \end{aligned}$$

**Bisection formula for $\lambda^\star$ and bounds.** We also get from Lemma 11 a bisection formula for $\lambda^\star$, which is a shorthand for $\lambda_{F[\partial\Omega]}^\star(y,\theta)$.

$$e^{-\lambda^\star} \sum_{i:y_i=0} e^{\theta_i} + \sum_{i:y_i>0} \frac{y_i}{W(y_i e^{-(\theta_i - \lambda^\star)})} = 1.$$

We focus here on a lower bound and an upper bound for $\lambda^\star \in \mathbb{R}$. Let us prove that

$$\log \sum_{i=1}^k e^{\theta_i} \leq \lambda^\star \leq \log 2 + \max\left\{ \log \sum_{i:y_i=0}^k e^{\theta_i}, \log \ell_0(y) + \max_{i:y_i>0} \theta_i + 2\ell_0(y)y_i \right\},$$

where $\ell_0(y) = \text{Card}(j : y_j \neq 0)$.

For the lower bound, we use the concavity [13] of the Lambert function $W$, which implies $\frac{1}{W(y_i e^{\lambda^\star - \theta_i})} \geq \frac{1}{y_i e^{\lambda^\star - \theta_i}}$. Thus,

$$1 \geq e^{-\lambda^\star} \sum_{i:y_i=0} e^{\theta_i} + \sum_{i:y_i>0} \frac{y_i}{y_i e^{\lambda^\star - \theta_i}},$$

which in turn implies

$$e^{\lambda^\star} \geq \sum_{i:y_i=0} e^{\theta_i} + \sum_{i:y_i>0} e^{\theta_i}$$

and yields the lower bound.

For the upper bound, the function $g(\lambda) = e^{-\lambda} \sum_{i:y_i=1} e^{\theta_i} + \sum_{i:y_i>0} \frac{y_i}{W(y_i e^{\lambda - \theta_i})}$ is continuous and decreasing (as it is a positive combination of decreasing functions) and $g(-\infty) = +\infty$. Thus if we find a $\lambda$ such that $g(\lambda) < 1$, we know that $\lambda^\star \leq \lambda$.

We deal with each term of $g(\lambda)$ separately. If $\lambda \in \mathbb{R}$ satisfies

$$e^{-\lambda} \sum_{i:y_i=0} e^{\theta_i} \leq \frac{1}{2}$$

$$\max_{i:y_i>0} \frac{y_i}{W(y_i e^{\lambda - \theta_i})} \leq \frac{1}{2\ell_0(y)},$$

then

$$g(\lambda) = \mathrm{e}^{-\lambda} \underbrace{\sum_{i:y_i=0} \mathrm{e}^{\theta_i}}_{\le 1/2} + \sum_{i:y_i>0} \underbrace{\frac{y_i}{W(y_i \mathrm{e}^{\lambda-\theta_i})}}_{\le 1/(2\ell_0(y))} \le 1.$$

Thus, all $\lambda$ satisfying the following inequalities are upper bounds of $\lambda^\star$

$$2 \sum_{i:y_i=0} \mathrm{e}^{\theta_i} \le \mathrm{e}^\lambda$$

$$2\ell_0(y)y_i \le W(y_i \mathrm{e}^{\lambda-\theta_i}), \forall i : y_i > 0.$$

As $W$ is monotone and $W^{-1}(t) = t\mathrm{e}^t$, we get

$$\log 2 + \log \sum_{i:y_i=0} \mathrm{e}^{\theta_i} \le \lambda$$

$$2\ell_0(y)\mathrm{e}^{2\ell_0(y)y_i} \le \mathrm{e}^{\lambda^\star - \theta_i}, \forall i : y_i > 0.$$

Thus taking $\lambda = \max \left\{ \log 2 + \log \sum_{i:y_i=0}^k \mathrm{e}^{\theta_i}, \max_{i:y_i>0} \log 2 + \log \ell_0(y) + \theta_i + 2\ell_0(y)y_i \right\}$ yields an upper bound of $\lambda^\star$.

## B.8 Proof of Proposition 7 (characterization of $F[\partial\Omega]$ using $D_\Omega$)

Let $(y, \theta) \in \operatorname{dom} \Omega \times \mathbb{R}^k$. We have

$$F[\partial\Omega](y, \theta) = \sup_{(y',\theta')\in\partial\Omega} \langle y - y', \theta' \rangle + \langle y', \theta \rangle$$

$$= \sup_{y'\in\operatorname{dom}\Omega} \left\{ \langle y', \theta \rangle + \sup_{\theta'\in\partial\Omega(y')} \langle y - y', \theta' \rangle \right\}$$

$$= \sup_{y'\in\operatorname{dom}\Omega} \left\{ \langle y', \theta \rangle - \Omega(y') + \Omega(y') + \sup_{\theta'\in\partial\Omega(y')} \langle y - y', \theta' \rangle \right\}$$

$$= \Omega(y) + \sup_{y'\in\operatorname{dom}\Omega} \langle y', \theta \rangle - \left( \Omega(y') + \Omega(y) - \Omega(y') - \sup_{\theta'\in\partial\Omega(y')} \langle y - y', \theta' \rangle \right)$$

$$= \Omega(y) + \sup_{y'\in\operatorname{dom}\Omega} \langle y', \theta \rangle - (\Omega(y') + D_\Omega(y, y'))$$

$$= \Omega(y) + (\Omega + D_\Omega(y, \cdot))^* (\theta)$$

$$= \Omega_y(y) + \Omega_y^*(\theta).$$

The supremum above is achieved at some $y' \in \partial\Omega_y^*(\theta)$.

When $\Omega = \Psi + \iota_\mathcal{C}$, where $\mathcal{C} \subseteq \operatorname{dom} \Psi$ is a nonempty closed convex set, using Lemmas 1 and 2, we have for all $y \in \mathcal{C}$

$$\Omega_y(y') = \Psi(y') + D_\Psi(y, y') + \iota_\mathcal{C}(y').$$

## B.9 Proof of Proposition 8 (lower bound)

It was shown in [8, Proposition 3] that if $f = g + \iota_\mathcal{C}$, where the function $g$ is Legendre-type with $\mathcal{C} \subseteq \operatorname{dom} \Psi$ a nonempty closed convex set, then for all $y \in \mathcal{C}$ and $\theta \in \mathbb{R}^k$,

$$0 \le D_g(y, \nabla f^*(\theta)) \le L_{f\oplus f^*}(y, \theta),$$

with equality if $\mathcal{C} = \operatorname{dom} g$. Using $g = \Psi_y$, $f = \Omega_y = \Psi_y + \iota_\mathcal{C}$, $y^\star = \nabla\Omega_y^*(\theta) = y_{F[\partial\Omega]}^\star(y, \theta)$, and Lemma 3, we therefore obtain

$$D_{\Psi_y}(y, y^\star) = \langle y - y^\star, \nabla^2\Psi(y^\star)(y - y^\star) \rangle \le L_{\Omega_y\oplus\Omega_y^*}(y, \theta) = L_{F[\partial\Omega]}(y, \theta).$$

If $\Psi_y$ is $\mu$-strongly convex and $D_\Psi$ is convex in its second argument, then $\Psi_y$ is $\mu$-strongly convex as well. Therefore, we also have

$$\frac{\mu}{2}\|y - y^\star\|_2^2 \le D_{\Psi_y}(y, y^\star).$$

# C   Comment on the inf-addition convention $+\infty + (-\infty) = +\infty$

The question of adding conflicting infinities has been thoroughly studied by Moreau in [23]. In this paper, Moreau introduced two additions for $\mathbb{R} \cup \{-\infty, +\infty\}$. One of Moreau's two conventions for summing infinities can be found under the name of inf-addition in [27, Chapter 1]. The other convention is named sup-addition.

Using the inf-addition on $\mathbb{R} \cup \{-\infty, +\infty\}$ allows, for instance, the equivalence of $\min_{x \in C} f(x)$ and $\min_{x \in \mathbb{R}^n} f(x) + \iota_C(x)$. The same goes for the sup-addition but for maximization problems.

We used the inf-addition convention for the definition of the generalized Bregman divergence in (1). As the generalized Bregman divergence is to be thought of as a generalization of a distance, it is generally a quantity that will be minimized. Therefore, the inf-addition is a good fit for the definition of the generalized Bregman divergence.

In the proof of Proposition 7 in B.8, the choice of inf-addition is corroborated by the fact that we want to calculate, for some fixed $y \in \mathbb{R}^k$, $\big(\Omega(\cdot) + D_\Omega(y, \cdot)\big)(\theta) = \sup_{y'} \langle y', \theta \rangle - \big(\Omega(y') + D_\Omega(y, y')\big), \forall y' \in \mathbb{R}^k$.

The minus sign transforms the inf-addition into the sup-addition when distributed. If we had chosen the sup-addition in the definition of the Bregman divergence $D_\Omega$, we would here calculate a supremum of an inf-addition, thus entering unknown territory.

