# OpenReview forum: "Learning with Fitzpatrick Losses"
_NeurIPS.cc/2024/Conference — NeurIPS 2024 poster_

### Official Review · Reviewer_3f1M · 2024-06-14

**Soundness:** 3
**Presentation:** 3
**Contribution:** 3
**Rating:** 7
**Confidence:** 3

**Summary:**

The paper proposes the Fitzpatrick loss, a tighter version of the Fenchel-Young loss. Fitzpatrick losses are defined based on the monotone operator theory. In particular, the Fitzpatrick function satisfies a tighter inequality than the Fenchel-Young inequality, leading to the tighter loss function. The authors elucidate the fundamental properties of the Fitzpatrick loss and present concrete examples, demonstrating that Fitzpatrick sparsemax/logistic losses indeed differ from the Fenchel-Young counterparts. The authors also show that the Fitzpatrick loss can be seen as the Fenchel-Young loss generated by target-dependent regularizes and that it is connected to generalized Bregman divergences, while the primal-dual nature of the Fitzpatrick loss crucially makes it convex. A lower bound on the Fitzpatrick loss is also presented, which is analogous to the Fenchel-Young counterpart. Finally, experiments compare Fitzpatrick and Fenchel-Young losses on classification tasks.

**Strengths:**

1. The paper presents a novel framework for designing loss functions from link functions. Prior to the work, the Fenchel-Young loss is the only such framework. The current work opens a new direction for designing various other loss functions with rigorous foundations of monotone operator theory. I believe this is of great benefit to the machine-learning community.
2. The paper discusses the properties of the Fitzpatrick loss and its connection to other notions (Fenchel-Young losses and generalized Bregman divergences) in detail. This is helpful for readers who are familiar with either of the existing notions.

**Weaknesses:**

A weakness (not so serious) is that the experimental results are not very compelling. Still, I find the slight superiority of Fitzpatrick-sparsemax to Sparsemax noteworthy as their computation costs are almost the same.

**Questions:**

1. A special form of target-dependent regularizers is used for constructing the cost-sensitive Fenchel-Young loss ([Blondel et al. 2020, Section 3.4](https://jmlr.csail.mit.edu/papers/v21/19-021.html)). Does the Fitzpatrick loss have any relation with the cost-sensitive Fenchel-Young loss?
2. Is it possible to establish calibration of target and Fitzpatrick excess risks, as in [Blondel (2019, Proposition 5)](https://proceedings.neurips.cc/paper_files/paper/2019/hash/7990ec44fcf3d7a0e5a2add28362213c-Abstract.html)? The lower bound in Proposition 8 seems useful for this.
3. Regarding lines 164-165, is there a canonical class of convex functions $\Omega$ such that $D\_\Omega$ is convex in the second argument?
4. In the experiments, predictions are produced by $\hat y\_\Omega(Wx)$, and I can understand this since the Fitzpatrick loss has $\hat y_\Omega$ as the link function as in Proposition 1. On the other hand, given the discussion in Section 3.3, $y^*\_{F[\partial\Omega]}(y, \theta)$ also appears to be a Fitzpatrick counterpart of $\hat y\_\Omega(\theta)$, which, however, cannot be used in the test phase since ground truth $y$ is not available. So, when $W$ is learned with the Fitzpatrick loss, aren't there other possible methods for converting $Wx$ to a prediction other than $\hat y\_\Omega(Wx)$ (without knowing $y$)?

#### **Minor comments**
- l. 84. "$\hat y\_\Omega$ is monotone..." Does this "monotone" refer to the one in Section 2.3? If so, it is not yet defined there.
- Eq. (4). Enclosing the terms after "argmax" in parentheses will make it easier to read.
- l. 174. that that.
- Proposition 8. Reminding that $\Psi_y$ is defined as with $\Omega_y$ in Section 3.3 is helpful; alternatively, it may be better to define $\Omega_y$ as a general notation in Section 3.3 or earlier.

**Limitations:**

The authors have discussed the limitation regarding computation costs.

---

> ### Author Rebuttal · Authors · 2024-08-06
>
> We thank the reviewer for the very positive and constructive feedback.
>
> > A special form of target-dependent regularizers is used for constructing the cost-sensitive Fenchel-Young loss (Blondel et al. 2020, Section 3.4). Does the Fitzpatrick loss have any relation with the cost-sensitive Fenchel-Young loss?
>
> There is similarity but the additional target-dependent term is linear in their case, while it is a general Bregman divergence in our case. We also tried to write the multiclass hinge loss as a Fitzpatrick loss, but we believe this is not possible.
>
> > Is it possible to establish calibration of target and Fitzpatrick excess risks, as in Blondel (2019, Proposition 5)? The lower bound in Proposition 8 seems useful for this.
>
> This is an interesting question but we leave the study of calibration guarantees to future work.
>
> > Regarding lines 164-165, is there a canonical class of convex functions
>  such that is convex in the second argument?
>
> We do not know a class of convex functions $\Omega$ such that the generated generalized Bregman distance $D_\Omega$ is convex in the second variable.
> We can see the difficulty of finding such convex functions by considering the case of differentiable $\Omega$ on the interior of its domain. In this case, we recover the classical Bregman divergence,
> $D_\Omega(y,y’) = \Omega(y) - \Omega(y’) - \langle y-y’,\nabla\Omega(y’) \rangle $.
> When $\Omega$ is the Shannon negentropy or the squared 2-norm, $D_\Omega(y,\cdot)$ is convex for all $y \in \mathrm{dom}~\Omega$ because
> $y’ \mapsto \langle y’, \nabla \Omega(y’) \rangle - \Omega(y’) $ is convex, which is usually
> not the case.
>
> > when is learned with the Fitzpatrick loss, aren't there other possible methods for converting $Wx$ to a prediction?
>
> It is indeed possible that other methods exist. For example, as an heuristic, we could try to replace the unknown $y$ with the average of the label proportions in the dataset. The calibration study could further suggest new principled methods.
>
> > l. 84. "is monotone..." Does this "monotone" refer to the one in Section 2.3? If so, it is not yet defined there.
>
> Thank you, we added a reference to section 2.3.
>
>  > Eq. (4). Enclosing the terms after "argmax" in parentheses will make it easier to read.
>
> Done, thanks.
>
> > Proposition 8. Reminding that  $\Psi_y$
>  is defined as with $\Omega_y$ in Section 3.3 is helpful; alternatively, it may be better to define
>  as a general notation in Section 3.3 or earlier.
>
> Done, thanks.

---

> > ### Comment · Reviewer_3f1M · 2024-08-12
> >
> > Thank you for the detailed answers to my questions. I retain my score of 7 (Accept).

---

### Official Review · Reviewer_C9fT · 2024-06-23

**Soundness:** 4
**Presentation:** 4
**Contribution:** 4
**Rating:** 9
**Confidence:** 4

**Summary:**

This article presents a refreshing take on losses used in ML, with solid motivational examples and a several of interesting and beautiful results in pure convex analysis.

**Strengths:**

The Fitzpatrick function is a well-known tool for achieving very pure-theoretical results like Minty's theorem in convex analysis. However, before reading this article, I was under the impression that the Fitzpatrick function was just a theoretical tool with limited applications in-practice...

Until now! This paper takes a beautiful tool from pure convex analysis, spruces up some very interesting and new results, and presents it as a new application in ML with some very solid justification concerning sharper inequalities in comparison to the classical class of Fenchel-Young inequalities.

**Weaknesses:**

My main regret is that, due to the reviewing load, I was not able to fully check all of the proofs presented in the appendix. The claims *in the article* are reasonable, interesting, well-presented, and thorough; however, I am quite concerned that there may be some (unintentionally) omitted hypothesis, or holes in the proofs in the appendix (e.g., please also see my statement about Compactness of the domain in "limitations" section), because this often happens in ML conference articles. I wish that an appendix counted as the "reviewed" part of the article, but alas I simply do not have the time to check every detail. I am going to suggest that this paper is accepted, but I suggest that the authors please take care to revise based on the following minor points.


Minor comments:
- Citations appear to be out of numerical order when they are cited
- Line 30: In what sense is "proper" meant? All losses I'm aware of
  are proper functions (using the convex analysis definition).
- Line 56: Is R_+ numbers which are strictly positive, or just
  positive?
- Line 59: The set must be nonempty for the projection to be defined.
- Line 62: This identity only holds when \Omega is differentiable  *and convex*! Otherwise, this definition of the subdifferential produces the emptyset in areas of concavity.
- Line 66: Please provide a citation.
- Line 68: Does this function need to be Legendre?
- Line 101: Monotone is defined here, but mentioned earlier in the
  article; would be nice to have the definition at the beginning.
- Definition 1: Within the text of the definition, "dom Omega"  appears before Omega is introduced.
- A bit more discussion would be nice on specifically which problems (4) can be calculated.
- Line 132: "Above expression" referring to "\nabla^2 Omega" is ambiguously defined, since in the constrained case Omega is not differentiable.

**Questions:**

- Line 69: As far as I've seen in most convex analysis books and papers, this convention is a bit nonstandard; typically, "+infinity +  (-infinity)" is *undefined*. Please comment on which results in this article break if the quantity "+infinity + (-infinity)" is undefined. I.e., if this is "not a valid move", which of your results still hold? It is very important to clarify which results hold under varying algebraic models of convex analysis.

- Line 157 / Prop. 7: the relationship between the claim in the equation following Line 157 and Proposition 7 is unclear. How does the claim in line 157 follow from Proposition 7?

**Limitations:**

The mathematically precise statements (propositions, lemmas, theorems/etc.) should state **all** of their assumptions. In particular, the authors do not repeat that a compact domain of the objective function is required for several of their results. (Unless I'm missing something and compactness is not actually required?)

The authors do a great job of explaining the results of their experiments. More numerics is always a plus, but since the article has so many strong theoretical contributions, I am quite happy with this as-is.

---

> ### Author Rebuttal · Authors · 2024-08-06
>
> We thank the reviewer for the very positive and detailed constructive feedback.
>
> > Citations appear to be out of numerical order when they are cited
>
> This is because the references are sorted alphabetically by author name.
>
> > Line 30: In what sense is "proper" meant? All losses I'm aware of are proper functions (using the convex analysis definition).
>
> In line 30, “proper” refers to “proper losses” or “proper scoring rules”, which we also mentioned in line 21 with citations (Gneiting et al. 2007) and (Grünwald et al. 2004). This notion of “proper” is not related to “proper functions” in convex analysis.
>
> > Line 56: Is R_+ numbers which are strictly positive, or just positive?
>
> We added $\mathbb{R}_+ := [0,+\infty)$ to the notation section.
>
> > Line 59: The set must be nonempty for the projection to be defined.
>
> Fixed.
>
> > Line 62: This identity only holds when \Omega is differentiable and convex! Otherwise, this definition of the subdifferential produces the emptyset in areas of concavity.
>
> Fixed.
>
> > Line 66: Please provide a citation.
>
> We added Proposition 11.3 from
> R. T. Rockafellar and R. J-B. Wets. Variational Analysis. Springer-Verlag, Berlin, 1998.
>
> > Line 68: Does this function need to be Legendre?
>
> Here, $\Omega$ needs not to be Legendre, as in Equation (1),
>  $\sup_{\theta’ \in \partial \Omega(y’)} \langle y-y’, \theta’ \rangle$ is always well defined, although it can take the value $-\infty$ or $\infty$. This is why we use the convention
> $+\infty +(-\infty) = +\infty$ in this definition.
>
> > Line 101: Monotone is defined here, but mentioned earlier in the article; would be nice to have the definition at the beginning.
>
> On line 84, we added a reference to section 2.3.
>
> > Definition 1: Within the text of the definition, "dom Omega" appears before Omega is introduced. A bit more discussion would be nice on specifically which problems (4) can be calculated.
>
> Fixed.
>
> > Line 132: "Above expression" referring to "\nabla^2 Omega" is ambiguously defined, since in the constrained case Omega is not differentiable.
>
> The reviewer is right, we have neglected to mention that the function is twice differentiable on the interior of its domain. This is now fixed.
>
> > Line 69: As far as I've seen in most convex analysis books and papers, this convention is a bit nonstandard; typically, "+infinity + (-infinity)" is undefined. Please comment on which results in this article break if the quantity "+infinity + (-infinity)" is undefined. I.e., if this is "not a valid move", which of your results still hold? It is very important to clarify which results hold under varying algebraic models of convex analysis.
>
> This question of adding conflicting infinites has been thoroughly studied by Moreau in:
>
> Inf-convolution, sous-additivité, convexité des fonctions numériques.
> J. Math. Pures Appl. (9), 49:109–154, 1970.
>
> In this paper, Moreau introduced two additions for $\mathbb{R} \cup \{-\infty,+\infty}$.
> One of the two Moreau’s conventions for summing the infinites can be found under the name of inf-addition in Chapter 1 Section E. in R. T. Rockafellar and R. J-B. Wets. Variational Analysis. Springer-Verlag, Berlin, 1998.
>
> To make a long story short, one of them is used for minimization and the other for maximization.
> Using the inf-addition on $\mathbb{R} \cup \{-\infty,+\infty}$ allows for things like $\min_C f$ coinciding with $\min_{\mathbb{R}^n} f(x) + \iota_C(x)$, where $C \subset \mathbb{R}^n$, and $f : \mathbb{R}^n \to \mathbb{R} \cup \{-\infty,+\infty}$.
>
> As the generalized Bregman divergence $D_\Omega$ is to be thought of as a generalization of a distance, it is generally a quantity that will be minimized. So, the inf-addition fits the definition of the generalized Bregman divergence in line 69.
>
> In the proof of Proposition 7, the choice of inf-addition is corroborated by the fact that we want to calculate, for some fixed $y\in \mathbb{R}^n$,
> $\big( \Omega(\cdot)+ D_\Omega(y,\cdot) \big)(\theta)
>  = \sup_{y’} \langle y’, \theta \rangle - \big( \Omega(y’)+ D_\Omega(y,y’) \big), \forall y’ \in \mathbb{R}^k$.
> The minus sign transforms the inf-addition into the sup-addition when distributed.
> If we had chosen the sup-addition in the definition of the Bregman divergence $D_\Omega$, we would here calculate a supremum of an inf-addition, thus entering unknown territory.
>
> To conclude, the natural choice for addition in the definition of the generalized Bregman divergence is the $+\infty + (-\infty) = + \infty$.
>
> > Line 157 / Prop. 7: the relationship between the claim in the equation following Line 157 and Proposition 7 is unclear. How does the claim in line 157 follow from Proposition 7?
>
> We are not sure if we understand the question correctly. Line 157 is a definition, not a claim. It is what we call the target-dependent regularization $\Omega_y$. With the definition in line 157, Proposition 7 shows that $F[\partial \Omega](y, \theta)$ coincides with $\Omega_y(y) + \Omega_y^*(\theta)$. And therefore the Fitzpatrick loss generated by $\Omega$ coincides with the Fenchel-Young loss generated by $\Omega_y$.

---

> > ### Comment · Reviewer_C9fT · 2024-08-13
> >
> > I thank the authors very much for their responses and clarifications.
> >
> > I appreciate the authors' motivation for using this model of defining $+\infty - (+\infty)$ as $+\infty$. I am aware of the Moreau article; however, other more modern convex analysis books (e.g., *Convex Analysis and Monotone Operator Theory in Hilbert Spaces* by Bauschke and Combettes) treat $+\infty - (+\infty)$ as an undefined term. This is an important distinction, since convex analysts certainly still use both notions in modern work. While I am still overall quite impressed with the article, I would like to see more emphasis on this distinction in the final version.
> >
> > Overall, it was a pleasure to review this article, and I look forward to seeing its final form in the future. I thank the authors for their time.

---

### Official Review · Reviewer_E7nS · 2024-07-10

**Soundness:** 3
**Presentation:** 3
**Contribution:** 3
**Rating:** 7
**Confidence:** 3

**Summary:**

This paper proposes a class of loss functions called Fitzpatrick loss, based on the Fitzpatrick function known in maximal monotone operator theory. It can be shown that building a loss function with the proposed method, can give a convex loss function that lower bounds the Fenchel-Young loss under the same choice of link function for prediction. Extensive analysis provides in the paper also shows the relationship between Fitzpatrick loss and Fenchel-Young loss. Finally, experiments were provided to validate the usefulness of Fitzpatrick loss.

**Strengths:**

1. The proposed method is theoretically sound and novel to the best of my knowledge. This give rises to a way to construct a convex loss function systematically that lower bounds the family of Fenchel-young loss under the same link function for prediction.
2. Not only the new class of loss function is proposed, but the characterization of the relationship between the proposed class of loss functions and existing loss was also provided to understand the research in this direction better.
3. The paper is well-written overall and is friendly for also new people in the field who is not familiar with Fenchel-young loss function.

**Weaknesses:**

Although it is nice to also provide the experiments in the paper, I found that the experiment failed to motivate the usefulness of Fitzpatrick loss family in the sense that the performance is not preferable overall. I think no so many insights were provided when Fitzpatrick loss is more useful than existing losses in the literature. The proposed loss is also more computationally demanding. Perhaps different kinds of experiments would give more values to the paper. From the current experiments, I think we still need lots of work to study when to use Fitzpatrick loss in practice.

My current score is Weak accept (6) as I put importance to the theory to advance the understanding of Fenchel-young loss and Fitspatrick loss function quite high and believe these are the main contributions of this paper, rather than the experimental result.

**Questions:**

1. Since Fitzpatrick loss lower bounds the Fenchel-young loss for the same link function, is it possible to find some use-cases that the proposed loss (e.g., Fitzpatrick-sparsemax) is preferable? For example, could it be more useful when we want to have sparse probability output and we have to learn under the case where there are outliers/noisy data?
2. It is said that we can provide a relationship between Fitzpatrick loss and Fenchel-young loss by using "target-dependent function" in Section 3.3 (Prop. 7). I think the result is very interesting and I have two questions regarding this finding.
- 2.1 Since we need target-dependent function for $\Omega_y$ to make the relationship holds, does the original Fenchel-Young loss supports target-dependent function case? If so, then this should be no problem and we can state that all properties of Fenchel-young loss inherits to Fitzpatrick loss.
- 2.2 Is it safe to say that Fitzpatrick loss is a special case of Fenchel-Young loss? Perhaps not?
- 2.3 Can we also say that any Fenchel-young loss can be rewritten in a form of Fitzpatrick loss generated by different $\Omega$?
3. In the conclusion section, there was a discussion about "there can only be one loss function associated with a certain link function". I am not sure about this statement for the definition of loss function here. Is it about a "proper composite" loss function, a "classification-calibrated" loss function, or a convex + (proper composite/classification-calibrated) loss function? Maybe we can be more explicit here otherwise one might think about any function or small ad-hoc modification using the same link and use it as a loss function with not much problems.


Minor comment
1. Typo: Line 99: also know as -> also known as
2. I feel it is a bit strange to say One loss is tighter than another loss since it is closer to zero everywhere.

**Limitations:**

The discussion was sufficient. The authors noted the challenge of computational problem and appropriately discussed the experimental results of the proposed method.

---

> ### Author Rebuttal · Authors · 2024-08-06
>
> We thank the reviewer for taking the time to review our paper and for the feedback.
>
> > Although it is nice to also provide the experiments in the paper, I found that the experiment failed to motivate the usefulness of Fitzpatrick loss family in the sense that the performance is not preferable overall.
>
> We believe it is important to provide empirical results, even if they are partially positive and partially negative. What is important is that our conclusions are backed up by observations. Based on our experiments on 11 datasets, our conclusions are:
> - FY-logistic performs better or comparably compared to FP-logistic and is computationally cheaper. Therefore, FY-logistic remains the solution of choice when we want to use the soft-argmax link.
> - FP-sparsemax performs better or comparably compared to FY-sparsemax and is computationally equivalent. Therefore, FP-sparsemax is a viable alternative to FY-sparsemax when we want to use the sparse-argmax link.
>
> > The proposed loss is also more computationally demanding.
>
> It is true that it is more computationally demanding to use the FP-logistic loss than its FY counterpart, as the former requires solving a root-finding problem (e.g., by bisection), while the latter enjoys a closed form (using the log-sum-exp).
>
> However, for FP-sparsemax, the situation is different. Indeed, computing the loss (and its gradient) is as computationally demanding as its FY counterpart (Proposition 5). Both only involve a projection on the simplex.
>
> > Since Fitzpatrick loss lower bounds the Fenchel-young loss for the same link function, is it possible to find some use-cases that the proposed loss (e.g., Fitzpatrick-sparsemax) is preferable? For example, could it be more useful when we want to have sparse probability output and we have to learn under the case where there are outliers/noisy data?
>
> Based on our experiments on 11 datasets, we found that FP-sparsemax is slightly better than FY-sparsemax. We also tried to add artificially various degrees of noise to labels but we did not observe any major change in our conclusions.
>
> > 2.1 Since we need target-dependent function for
>  to make the relationship holds, does the original Fenchel-Young loss supports target-dependent function case? If so, then this should be no problem and we can state that all properties of Fenchel-young loss inherits to Fitzpatrick loss.
>
> In principle, Fenchel-Young losses do not support target-dependent regularization functions $\Omega_y$. The problem comes at prediction time, as we cannot compute $\nabla \Omega_y^*$, since $y$ is unknown. However, viewing Fitzpatrick losses as Fenchel-Young losses with target-dependent $\Omega_y$ is useful because, as you point out and as we write in line 162, properties can be inherited easily from Fenchel-Young losses. For example, if $\Omega_y$ is strongly convex, then the Fitzpatrick loss associated with $\Omega$ is smooth.
>
> > 2.2 Is it safe to say that Fitzpatrick loss is a special case of Fenchel-Young loss? Perhaps not?
>
> The answer is no by the theory of representations of maximal monotone operators (Burachik et al., 2002).
> However, in the paper we introduce the notion of target dependent Fenchel-Young losses (which are not strictly speaking Fenchel-Young losses). Indeed, one of the contributions of our paper is to rewrite Fitzpatrick losses as Fenchel-Young losses with a target-dependent $\Omega_y$ (Proposition 7), while keeping the same target-independent link function $\nabla \Omega^*$.
>
> > 2.3 Can we also say that any Fenchel-young loss can be rewritten in a form of Fitzpatrick loss generated by different  $\Omega$?
>
> As both Fenchel-Young and Fitzpatrick losses only depend on the subdifferential of the generating function, the question boils down to ask if a Fenchel-Young loss generated by some subdifferential can be rewritten in the form of a Fitzpatrick loss for a different subdifferential.
> The answer is no by Theorem 6.1 in (Burachik et al., 2002).
>
> > In the conclusion section, there was a discussion about "there can only be one loss function associated with a certain link function". I am not sure about this statement for the definition of loss function here. Is it about a "proper composite" loss function, a "classification-calibrated" loss function, or a convex + (proper composite/classification-calibrated) loss function? Maybe we can be more explicit here otherwise one might think about any function or small ad-hoc modification using the same link and use it as a loss function with not much problems.
>
> Thank you for catching. We indeed meant convex loss function. Otherwise, it’s indeed possible to compose the link with some other loss function but the resulting composite loss function would be nonconvex in general. We also omit trivial loss modifications such as multiplying the loss by a non-negative scalar.
>
> We fixed the typos reported by the reviewer.

---

> > ### Comment · Reviewer_E7nS · 2024-08-12
> > **Thank you for the author feedback**
> >
> > I have read other reviews as well as the author rebuttal. Thank you very much for the detailed feedback.
> >
> > Thank you for clarifying my concerns and my potential misunderstanding. I agree with other reviewers and the paper that this paper provides a theoretical foundation for the Fitzpatrick loss which is novel to the best of my knowledge. Even though experimental results are not very strong and thus more investigation of empirical performance is needed, it can be studied in the future for some specific applications of interest.
> >
> > In this current form, I think the paper is worth an acceptance. As a result, I raised the score to 7 (Accept).

---

### Official Review · Reviewer_26BT · 2024-07-13

**Soundness:** 3
**Presentation:** 3
**Contribution:** 2
**Rating:** 6
**Confidence:** 3

**Summary:**

The paper describes an alternative approach to constructing loss functions. The authors propose a Fitzpatrick loss and compare it to Fenchel-Young loss and Bregman divergence. The work enumerates the basic properties of the introduced loss and compares it to existing loss functions by numerical simulations.

**Strengths:**

The overview of Fitzpatrick loss is nice. For the reader not acknowledged with the topic it is easy to get the main concepts. The authors also give examples and compare Fitzpatrick loss with Fenchel-Young loss.

**Weaknesses:**

I think the paper needs more thorough numerical evaluation due to the format of the conference. Although the properties of Fitzpatrick loss are given, the logical question is, why to use it instead of known loss functions for machine learning? At the moment, the authors presented comparison for classification problem over several datasets. Without broader experimental study, the manuscript seems more like a mathematical note. Therefore, I think that additional experiments will strengthen the paper.

Moreover, the datasets are for classification, but the results are compared in terms of mean squared error. It seems more logical to compare using some metric for classification, i.e. accuracy, f1-score etc.

**Questions:**

The paper claims that the Fitzpatrick losses are tighter than Fenchel-Young. Can something be said from the statistical point of view? Does using new loss lead to better generalization, for example.

**Limitations:**

The paper does not have negative societal impact.

---

> ### Author Rebuttal · Authors · 2024-08-06
>
> We thank the reviewer for taking the time to review our paper and for the feedback.
>
> > I think the paper needs more thorough numerical evaluation due to the format of the conference.
>
> Our paper is first and foremost a theoretical contribution:
> - it introduces a new family of losses that lower-bound Fenchel-Young losses;
> - it is the first practical application of the Fitzpatrick function;
> - it advances the Fitzpatrick function literature, e.g., Proposition 7 is a novel result.
>
> That said, based on our empirical comparison on **11 datasets**, our main claim is that FP-sparsemax is a viable alternative to FY-sparsemax, as it performs better or comparably, and is not more computationally demanding.
>
> > Moreover, the datasets are for classification, but the results are compared in terms of mean squared error. It seems more logical to compare using some metric for classification, i.e. accuracy, f1-score etc.
>
> Following Blondel et al (2020), we are doing label proportion estimation experiments. Since the model outputs are probability distributions over the classes, the mean squared error is an appropriate measure for estimating the quality of prediction. It is known as the **Brier score**, in a probabilistic forecasting context.
>
> > Can something be said from the statistical point of view? Does using new loss lead to better generalization
>
> We leave additional theoretical guarantees such as calibration or generalization to future work, as we believe the paper is already quite packed with results (8 propositions in the main text).

---

> > ### Comment · Reviewer_26BT · 2024-08-09
> > **Answer to Rebuttal**
> >
> > Dear Authors,
> >
> > Thank you for addressing my comments. I decided to raise my score by 1 point.

---

### Official Review · Reviewer_xpBU · 2024-07-13

**Soundness:** 4
**Presentation:** 3
**Contribution:** 2
**Rating:** 5
**Confidence:** 3

**Summary:**

The paper proposes a family of losses based on the notion of a Fitzpatrick function, which takes the role of composable subdifferentials in Fenchel-Young losses. The resulting family of losses is parallel to Fenchel-Young losses in the sense that each Fenchel-Young loss has a shared link function with a certain Fitzpatrick loss, but the losses themselves differ. It is shown that these losses posses some desirable properties such as convexity, and that they are based on a type of duality that yields a smaller duality gap than the one based on the Fenchel-Young inequality. Representative Fitzpatrick losses, such as logistic and sparsemax losses are developed as examples of useful members of the family.

Experiments show that the losses are mostly comparable in performance to their Fenchel-Young parallels, with the exception of the sparsemax loss that might be slightly improved in its Fitzpatrick version.

**Strengths:**

The approach is original and imports ideas from operator theory into ML. Theory is laid out neatly and in an easy to understand manner. Loss functions are clearly an important part of training any ML model, hence improvements in them can be very significant even if they are small, and I assume that further progress beyond this paper can be significant in practice as well.

**Weaknesses:**

The empirical utility of the developed losses in not entirely clear, since they do not offer a significant advantage over standard losses, and in some cases (e.g. Fitzpatrick sigmoid) may be more computationally demanding.
In other losses such as the squared loss and hinge loss, the parallel Fitzpatrick losses are practically the same. Hence, the contribution here is mainly in exploring further definitions of losses that are associated with familiar link functions, but attach to them a loss function that is not the standard one that stems from the Fenchel-Young family.

I am not an expert on this topic, so while I'm unsure whether the paper is interesting for its sub-community within ML, or for the wider NeurIPS audience (mainly because readers put a lot of emphasis on empirical results), I found it interesting enough to read hence I will give a borderline acceptance rating.

**Questions:**

No significant questions arise from reading the paper. It might be nice if the authors could comment on what potential they see for future research on this topic, and how

**Limitations:**

The authors mention some limitations such as somewhat increased computation time, and acknowledge the mixed empirical results. I find this to be an adequate treatment of limitations.

---

> ### Author Rebuttal · Authors · 2024-08-06
>
> We thank the reviewer for taking the time to review our paper and for the feedback.
>
> > The empirical utility of the developed losses in not entirely clear, since they do not offer a significant advantage over standard losses, and in some cases (e.g. Fitzpatrick sigmoid) may be more computationally demanding.
>
> Indeed, as we clearly acknowledged in the experiment section, FP-logistic does not bring much benefit compared to FY-logistic. However, FP-sparsemax brings benefits compared to FY-sparsemax: FP-sparsemax performs better than FY-sparsemax for some datasets and comparably for the others. Therefore, FP-sparsemax should be considered as a viable alternative to FY-sparsemax, as it is not more computationally demanding. Indeed, as explained in Proposition 5 and in the text below it, FP-sparsemax is based on the projection on the simplex, just like FY-sparsemax.
>
> Furthermore, the FP-sparsemax loss can also be trained by dual coordinate ascent, similarly to the FY-sparsemax loss (see Section 8.2, Blondel et al. 2020). Indeed, for $\Omega(y) = \frac{1}{2}\lVert y \rVert^2 + \iota_{\triangle^k}(y)$,  dual coordinate ascent consists of computing these proximal operators for each subproblem associated with the sample $i$:
> - for FY-sparsemax: $\mathrm{prox}_{\frac{1}{\sigma_i} \Omega}(v_i)$;
> - for FP-sparsemax: $\mathrm{prox}_{\frac{2}{\sigma_i} \Omega}(\frac{v_i - y_i}{\sigma_i})$.
>
> Both $\mathrm{prox}$ have a similar computational cost, as they are both computed using formulas for $\mathrm{prox}_{\tau \Omega}(\eta)$, where $\tau \in \mathbb{R}$ and $\eta \in \mathbb{R}^k$, (see Section 8.4, Blondel et al. 2020).
>
> Therefore, FP-sparsemax and FY-sparsemax are equivalent in terms of computational cost, whether we use primal or dual training.
>
> > It might be nice if the authors could comment on what potential they see for future research on this topic
>
> Future directions include calibration guarantees, new link functions and more efficient training for the FP-logistic loss.

---

> > ### Comment · Reviewer_xpBU · 2024-08-11
> > **Post Rebuttal Update**
> >
> > Thank you for the response.
> >
> > I understand that the point of the paper is not to introduce "better" losses than FY, and appreciate the paper for the results it presents. Therefore I will keep my recommendation to accept the paper.

---

### Author Rebuttal · Authors · 2024-08-06

We thank the reviewers for the very positive and constructive feedback, as well as the ACs for their editorial work. We summarize below our main replies to the reviewers (see reply to each reviewer for more details).

* All comments have been addressed (minor typos, some assumptions that were in the text have been moved explicitly in the propositions).
* Our paper is first and foremost a theoretical contribution (**8 propositions** in the main text), advancing both the literature on loss functions and the literature on Fitzpatrick functions. It is the first practical application of the Fitzpatrick function.
* Following Blondel et al (2020), we are doing label proportion estimation experiments and we use MSE (also known as **Brier score**) for evaluation.
* Based on our experiments on **11 datasets**, our conclusions are:
  - FY-logistic performs better or comparably compared to FP-logistic and is computationally cheaper. Therefore, FY-logistic remains the solution of choice when we want to use the soft-argmax link (a.k.a. softmax).
  - FP-sparsemax performs better or comparably compared to FY-sparsemax and is computationally equivalent. Therefore, FP-sparsemax is a viable alternative to FY-sparsemax when we want to use the sparse-argmax link (a.k.a. sparsemax).

---

### Decision · Program_Chairs · 2024-09-25

**Decision:**

Accept (poster)

**Comment:**

Overall, all reviewers found the main theoretical contribution of this paper (the introduction of the convex Fitzpatrick loss functions) to be compelling and well explained. It is thus an easy accept. Most reviewers found empirical motivation a bit lacking, since the new loss functions do not seem to lead to any especially large performance gains in common settings. However, this was mitigated by the strong theoretical contributions of the work. Hopefully the paper will spur further investigation on the empirical side of things.